# Single-cell transcriptomics reveals a new dynamical function of transcription factors during embryonic hematopoiesis

Isabelle Bergiers[1†], Tallulah Andrews[2†], Özge Vargel Bölükbaşı[1†‡], Andreas Buness[1], Ewa Janosz[1§], Natalia Lopez-Anguita[1], Kerstin Ganter[1], Kinga Kosim[1#], Cemre Celen[1¶], Gülce Itır Perçin[1**], Paul Collier[3††], Bianka Baying[3], Vladimir Benes[3], Martin Hemberg[2*], Christophe Lancrin[1*]

[1]European Molecular Biology Laboratory, EMBL Rome, Monterotondo, Italy; [2]Wellcome Trust Sanger Institute, Hinxton, United Kingdom; [3]Genomics Core Facility, European Molecular Biology Laboratory, Heidelberg, Germany

**\*For correspondence:**
mh26@sanger.ac.uk (MH);
christophe.lancrin@embl.it (CL)

[†]These authors contributed equally to this work

**Present address:** [‡]Boston's Children Hospital, Boston, United States; [§]Institute of Experimental Hematology, Hannover, Germany; [#]University of Sheffield, Sheffield, United Kingdom; [¶]UT Southwestern Medical Center, Dallas, United States; [**]Technische Universität Dresden, Dresden, Germany; [††]Weill Cornell Medicine, New York, United States

**Competing interests:** The authors declare that no competing interests exist.

**Abstract** Recent advances in single-cell transcriptomics techniques have opened the door to the study of gene regulatory networks (GRNs) at the single-cell level. Here, we studied the GRNs controlling the emergence of hematopoietic stem and progenitor cells from mouse embryonic endothelium using a combination of single-cell transcriptome assays. We found that a heptad of transcription factors (Runx1, Gata2, Tal1, Fli1, Lyl1, Erg and Lmo2) is specifically co-expressed in an intermediate population expressing both endothelial and hematopoietic markers. Within the heptad, we identified two sets of factors of opposing functions: one (Erg/Fli1) promoting the endothelial cell fate, the other (Runx1/Gata2) promoting the hematopoietic fate. Surprisingly, our data suggest that even though Fli1 initially supports the endothelial cell fate, it acquires a pro-hematopoietic role when co-expressed with Runx1. This work demonstrates the power of single-cell RNA-sequencing for characterizing complex transcription factor dynamics.
DOI: https://doi.org/10.7554/eLife.29312.001

## Introduction

Over the past decade, advances in high-throughput transcriptomics and DNA occupancy analyses have provided valuable insights into how transcription factors (TFs) regulate cell fate decisions and determine cell identity. It has become clear that during embryonic development, the expression of key TFs is strictly regulated, and it has been shown that the combinatorial expression of a relatively limited number of TFs is sufficient to establish (and potentially change) cell identity and differentiation through their action on the underlying gene regulatory networks (GRNs) (*Iwafuchi-Doi and Zaret, 2016*). In this context, bulk transcriptomics analysis and chromatin immuno-precipitation sequencing (ChIP-seq) were used to describe TF interactions, thereby providing GRN models (*Dunn et al., 2014*; *Mullen et al., 2011*). Recently, these techniques were successfully used to study the embryonic hematopoietic system in an in vitro embryonic stem cell (ESC) differentiation model (*Goode et al., 2016*).

However, the use of bulk cell populations constitutes an important limitation in our efforts to fully understand the GRNs. Although bulk transcriptomics can reveal crucial overall gene correlations between semi-stable cellular states, it cannot resolve subtler gene interactions occurring in complex transitional states. In addition, using a bulk approach makes it difficult to infer the direct consequences on the transcriptional landscape upon which these TFs are acting.

These limitations can be overcome by the use of single-cell approaches. Over the past five years, tremendous technical progress has been achieved in the field. Gene expression can be efficiently

assessed at the single-cell level, making it possible to distinguish subpopulations within tissues and cell cultures (*Kolodziejczyk et al., 2015*; *Scialdone et al., 2016*). Single-cell transcriptomics has previously been used to unravel complex developmental transitions such as gastrulation (*Scialdone et al., 2016*), demonstrating that it is possible to determine combinations of TFs that are expressed at the single-cell level as cellular differentiation progresses.

In the present work, we applied single-cell transcriptomics approaches to the ontogeny of the blood system in the mouse embryo. The GRNs that are involved in the production of the hematopoietic stem and progenitor cells (HSPCs) at the origin of all blood cells are not well understood. During mouse embryogenesis, HSPCs are generated mainly in the yolk sac (YS), from E7–E7.5, and in the aorta-gonad-mesonephros (AGM) region, from E10.5. They emerge from the endothelial cells forming the vasculature, through a process called the endothelial to hematopoietic transition (EHT) (*Boisset et al., 2010*; *Choi et al., 1998*; *Eilken et al., 2009*; *Lancrin et al., 2009*; *Zovein et al., 2008*). It has been proposed that a heptad of seven TFs (Gata2, Runx1, Erg, Fli1, Lmo2, Lyl1 and Tal1) known to be essential for blood development form a transcriptional complex that is potentially involved in the generation of HSPCs. This proposal was based on bulk ChIP-seq analysis of the binding of these seven TFs to the regulatory elements of 927 genes in the HPC7 cell line (*Wilson et al., 2010*). Nonetheless, there was no direct evidence that all seven TFs were expressed together at the single-cell level during embryogenesis or that the heptad targets play a crucial role in development. Employing single-cell transcriptomics analysis and an in vitro ESC differentiation system that gives rise to blood cells, we addressed these questions and provided novel insights into the formation of HSPCs from endothelial cells. Our data show that during EHT, two sets of TFs have opposite effects that allow proper differentiation. Through the transition, Erg and Fli1 support the endothelial cell fate while Runx1 and Gata2 promote the hematopoietic one. Unexpectedly, we found that the endothelial transcription factor Fli1 could acquire a hematopoietic function when expressed together with the hematopoietic master regulator Runx1. This work shows that GRN analysis based on single-cell transcriptomics data can highlight biological aspects that are missed by classical bulk methods, emphasizing the power of single-cell approaches in the understanding of complex developmental transitions.

## Results

### Key transcription factors identify a population intermediary between endothelial and blood cells during EHT in vivo and in vitro

We performed single-cell quantitative RT-PCR (sc-q-RT-PCR) on 95 genes associated with hematopoietic, endothelial (Endo), and vascular smooth muscle (VSM) cells (*Supplementary file 1*). Single cells were sorted from YS and AGM dissected from mouse embryos at E9, E10.5 and E11. To enrich for cells undergoing EHT, we selected cells using both the endothelial marker VE-cadherin (VE-Cad) and the hematopoietic marker CD41 (*Figure 1—figure supplements 1* and *2*). Using hierarchical clustering, these cells could be separated into three major groups in both tissues: Endo cells, Pre-HSPCs, which expressed both hematopoietic and endothelial genes (*Taoudi et al., 2008*), and HSPCs (*Figure 1A and B*, *Figure 1—figure supplements 1* and *2*).

While the frequency of each population of cells differed between tissues and time-points (*Figure 1—figure supplement 3*), groups were more strongly correlated (Pearson r > 0.9) to their counterparts than to other groups in the same tissue and clustered together using principal component analysis (PCA) and hierarchical clustering (*Figure 1—figure supplement 4* and *Figure 1A*).

In addition, E9.5 YS $Gfi1^{-/-}Gfi1b^{-/-}$ cells, which are incapable of completing EHT due to the lack of the two transcriptional repressors Gfi1 and Gfi1b (*Lancrin et al., 2012*), clustered together with the Pre-HSPCs, reinforcing the notion that this population is an essential intermediary step between Endo and HSPCs (*Figure 1—figure supplement 5* and *Figure 1B*).

EHT can be recapitulated in vitro using the ESC differentiation model (*Huber, 2010*). We performed a 3.25 day differentiation followed by isolation of Flk1$^+$ cells, enriched in blast colony forming cells (BL-CFCs) (*Choi et al., 1998*), which were cultured for a further 1.5 days before performing sc-q-RT-PCR using the same 95 genes (*Figure 1C*). Hierarchical clustering identified four groups, corresponding to Endo, HSPCs, and Pre-HSPCs found in the embryonic vasculature and a fourth group characterised by high expression of *Acta2* (smooth muscle actin) and *Serpine1* and by low expression

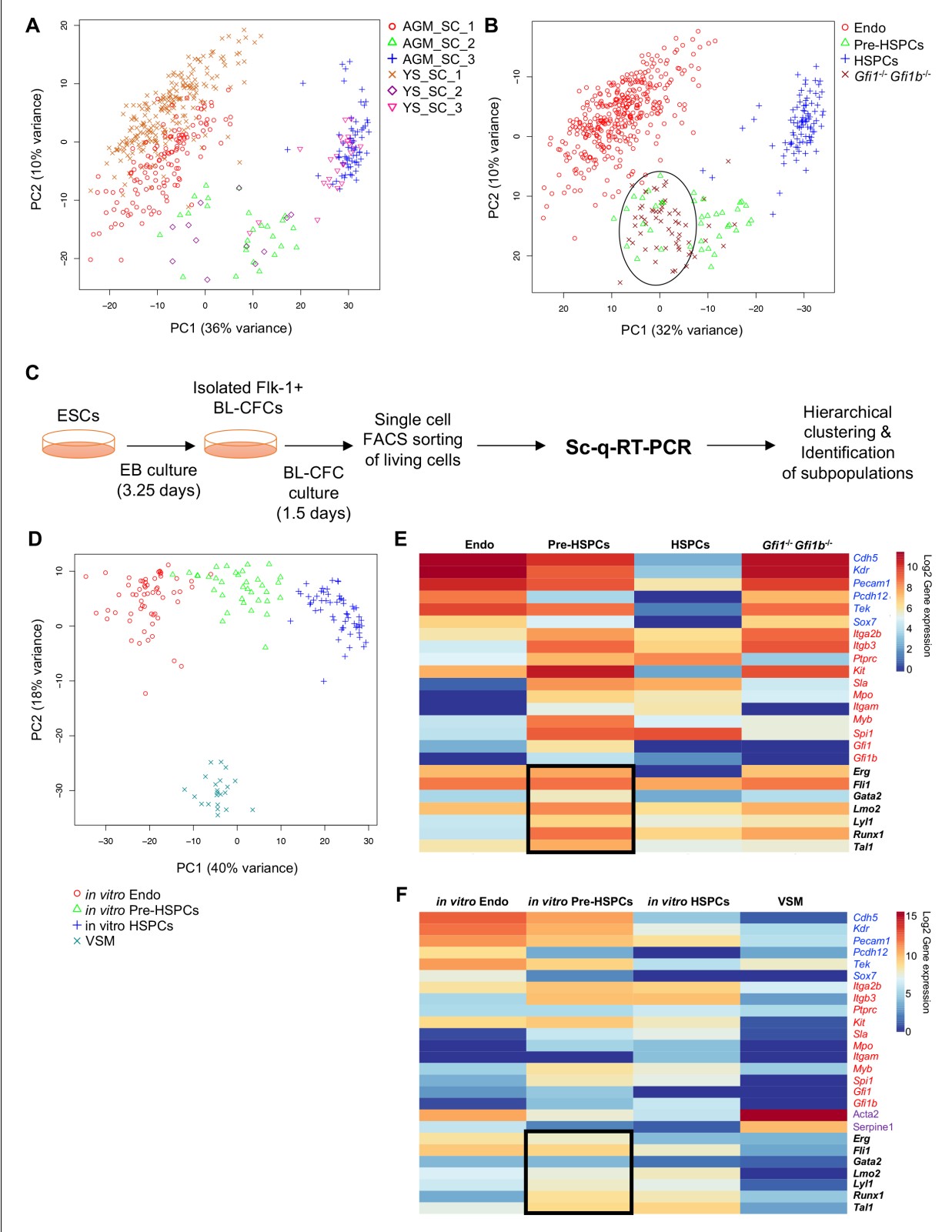

**Figure 1.** Analysis of in vivo and in vitro EHT by single-cell qRT-PCR identified the expression of key TFs at the Pre-HSPC stage. (A) Principal component analysis (PCA) plot showing the cells isolated at E9, E10.5 and E11 from AGM and YS according to the indicated sample clusters (SC). For each tissue, each time point represents one experiment. (B) PCA plot showing the *Gfi1⁻ᐟ⁻ Gfi1b⁻ᐟ⁻* GFP⁺ YS cells mixing with Pre-HSPCs group (black ellipse). Note that the PC2 axis has been reversed. (C) Experimental workflow used to differentiate in vitro ESCs into blood cells. BL-CFC, blast colony

*Figure 1 continued on next page*

*Figure 1 continued*

forming cells; EB, embryoid body; FACS, fluorescence-activated cell sorting. (**D**) PCA plot showing the four groups of cells coming from in vitro differentiated ESCs. (**E**) Heatmap showing average expression of endothelial (blue), hematopoietic genes (red) and seven key TFs (black) in the four groups found in vivo. The black rectangle highlights the expression of the seven TFs. (**F**) Heatmap showing the average expression of endothelial (blue), hematopoietic (red), vascular smooth muscle genes (purple) and seven key TFs (black) in the four groups found after in vitro differentiation of ESCs. The black rectangle highlights the expression of the seven TFs. See also *Figure 1—figure supplements 1–6*, *Supplementary file 1*, *2*, *3* and *11*.

DOI: https://doi.org/10.7554/eLife.29312.002

The following figure supplements are available for figure 1:

**Figure supplement 1.** Hierarchical clustering analysis of AGM VE-Cad+ cells.

DOI: https://doi.org/10.7554/eLife.29312.003

**Figure supplement 2.** Hierarchical clustering analysis of YS VE-Cad+ cells.

DOI: https://doi.org/10.7554/eLife.29312.004

**Figure supplement 3.** Time-course analysis of EHT in the AGM and YS.

DOI: https://doi.org/10.7554/eLife.29312.005

**Figure supplement 4.** Comparison of EHT subpopulations from AGM and YS.

DOI: https://doi.org/10.7554/eLife.29312.006

**Figure supplement 5.** Hierarchical clustering analysis of YS Gfi1$^{-/-}$ Gfi1b$^{-/-}$ cells.

DOI: https://doi.org/10.7554/eLife.29312.007

**Figure supplement 6.** Hierarchical clustering analysis of in vitro differentiated ESCs.

DOI: https://doi.org/10.7554/eLife.29312.008

of endothelial and hematopoietic genes, which we identified as VSM (*Figure 1—figure supplement 6A* and *Figure 1D*).

In both the in vivo and in vitro systems, the most notable characteristic of the Pre-HSPCs population is the co-expression of seven genes coding for hematopoiesis-associated transcription factors: *Erg*, *Fli1*, *Tal1*, *Lyl1*, *Lmo2*, *Runx1* and *Gata2* (*Figure 1E and F*, *Figure 1—figure supplement 6B and C*), which have been proposed to work together as a complex (*Wilson et al., 2010*). By contrast, Endo cells expressed only *Erg*, *Fli1*, *Lmo2* and *Tal1* whereas HSPCs expressed *Fli1*, *Lmo2*, *Lyl1*, *Runx1* and *Tal1*. This suggested that these seven TFs might be important in establishing and/or maintaining the cell-type identity of the Pre-HSPCs population.

## Simultaneous overexpression of Erg, Fli1, Tal1, Lyl1, Lmo2, Runx1, Cbfb and Gata2 during hemangioblast differentiation leads to the formation of a population resembling Pre-HSPCs

The co-expression of the seven TFs in the Pre-HSPCs populations in vivo and in vitro prompted us to ask whether the co-expression of these genes could be linked to the identity of the Pre-HSPCs. To determine the effect of the expression of these seven factors at the single-cell level, we established an inducible ES cell line in which all seven genes and *Cbfb*, a protein that is essential for Runx1 DNA binding and function (*Wang et al., 1996*; *Tahirov et al., 2001*), could be expressed simultaneously following the addition of doxycycline (dox). We generated an inducible ES cell line (i8TFs) in which the eight coding sequences were linked together by T2A sequences (*Figure 2A*). This strategy exploits the 'ribosomal skipping' mechanism of the viral T2A peptide (*Donnelly et al., 2001*) to allow the production of eight proteins from a single transcript. The expression of all eight proteins was validated by western blot after 24 hr of dox treatment in ESC culture (*Figure 2B*). As a control, we created an ESC line, which does not contain any novel cDNA (Empty), but is otherwise identical to the i8TFs line.

We studied the effect of the overexpression of the eight TFs during hemangioblast differentiation. After three days of EB culture, the Flk1$^+$ cells were differentiated for one day in BL-CFC culture, at which time the cells have lost their mesoderm identity and have produced endothelial and vascular smooth muscle cells. Dox was added at this time point, and cells were cultured for two more days. Cells were imaged every 15 min for 48 hr and were harvested at the end of the culture for flow cytometry. The Empty cell line did not show any major difference, neither in terms of cell surface markers nor in terms of cell morphology after dox treatment. By contrast, we noticed a dramatic change following the activation of the eight TFs when almost all cells developed the same phenotype (VE-Cad$^+$ CD41$^+$ and cKit$^+$ CD41$^+$ depending on the staining) (*Figure 3A*). This change was

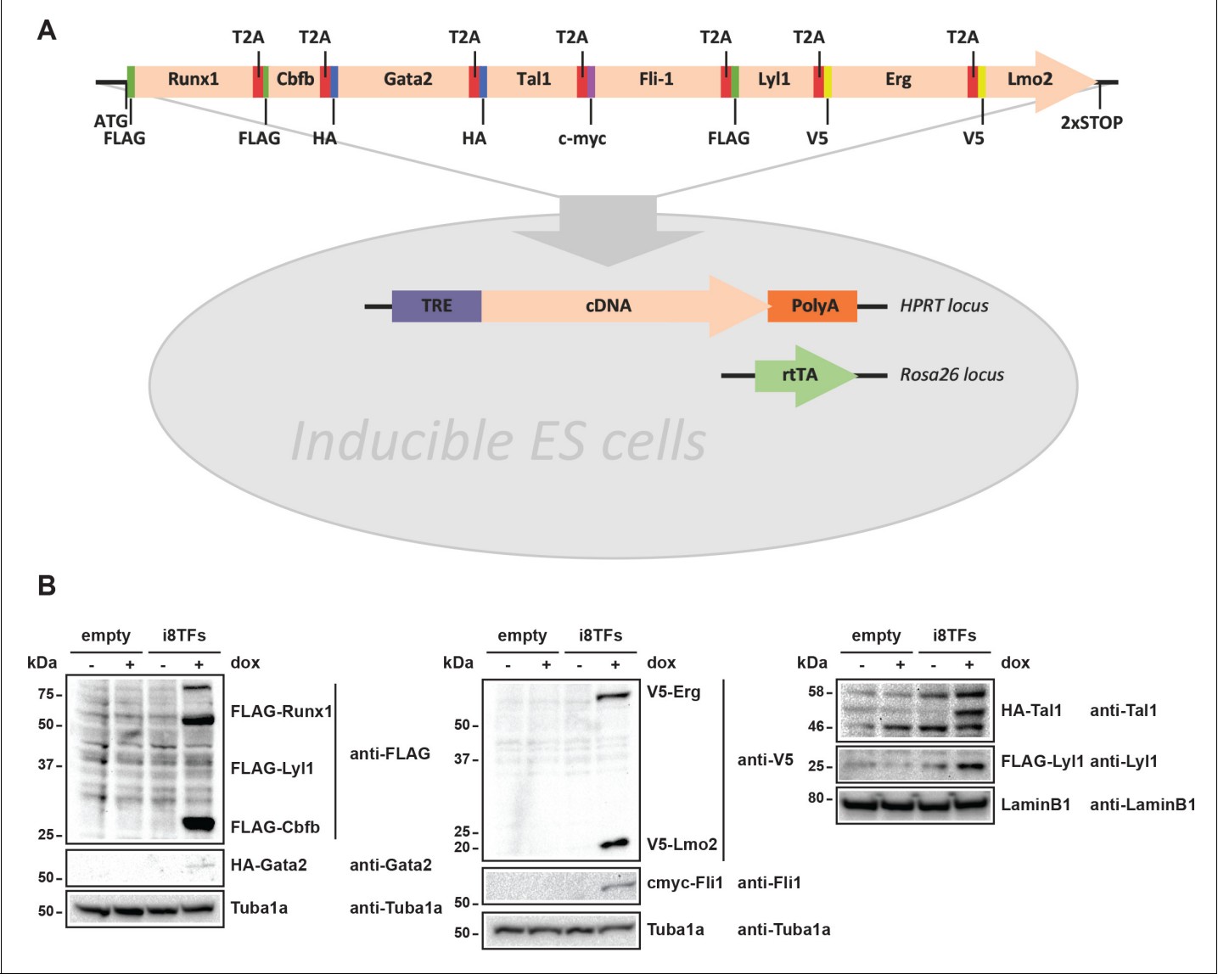

**Figure 2.** Generation of a doxycycline-inducible ESC line for the simultaneous expression of the eight transcription factors. (**A**) Scheme showing the generation of the i8TFs ESC line. (**B**) Western blot showing the protein expression of the eight TFs after doxycycline (dox) treatment in the Empty line and in the i8TFs ESC line. See also **Supplementary file 11**.

DOI: https://doi.org/10.7554/eLife.29312.009

accompanied by a marked decrease in the number of round cells (**Figure 3B** and **Figure 3—figure supplement 1**). The simultaneous expression of the eight TFs did not change the frequency of living cells (**Figure 3C**) but there was a 40% decrease in the number of cells in S phase, suggesting that the expression of the eight TFs has a negative impact on cell proliferation (**Figure 3D**).

To evaluate the impact of the simultaneous overexpression of these eight genes on hematopoietic differentiation activity, we performed a colony-forming unit (CFU) assay. We plated the cells from both the i8TFs – dox and the i8TFs + dox conditions in the absence of dox in a medium that allows the growth of both myeloid and erythroid cells. We observed a dramatic increase in the number of colony-forming units from the +dox cells, suggesting that the induction of the eight TFs increases hematopoietic differentiation potential (**Figure 3E**).

The overexpression of the eight TFs led to the production of a majority of VE-Cad$^+$ CD41$^+$ cells (**Figure 3A**). However, at the beginning of dox treatment, the majority of cells were VE-Cad$^-$ CD41$^-$ (**Figure 3—figure supplement 2A**), which were enriched in vascular smooth muscle cells

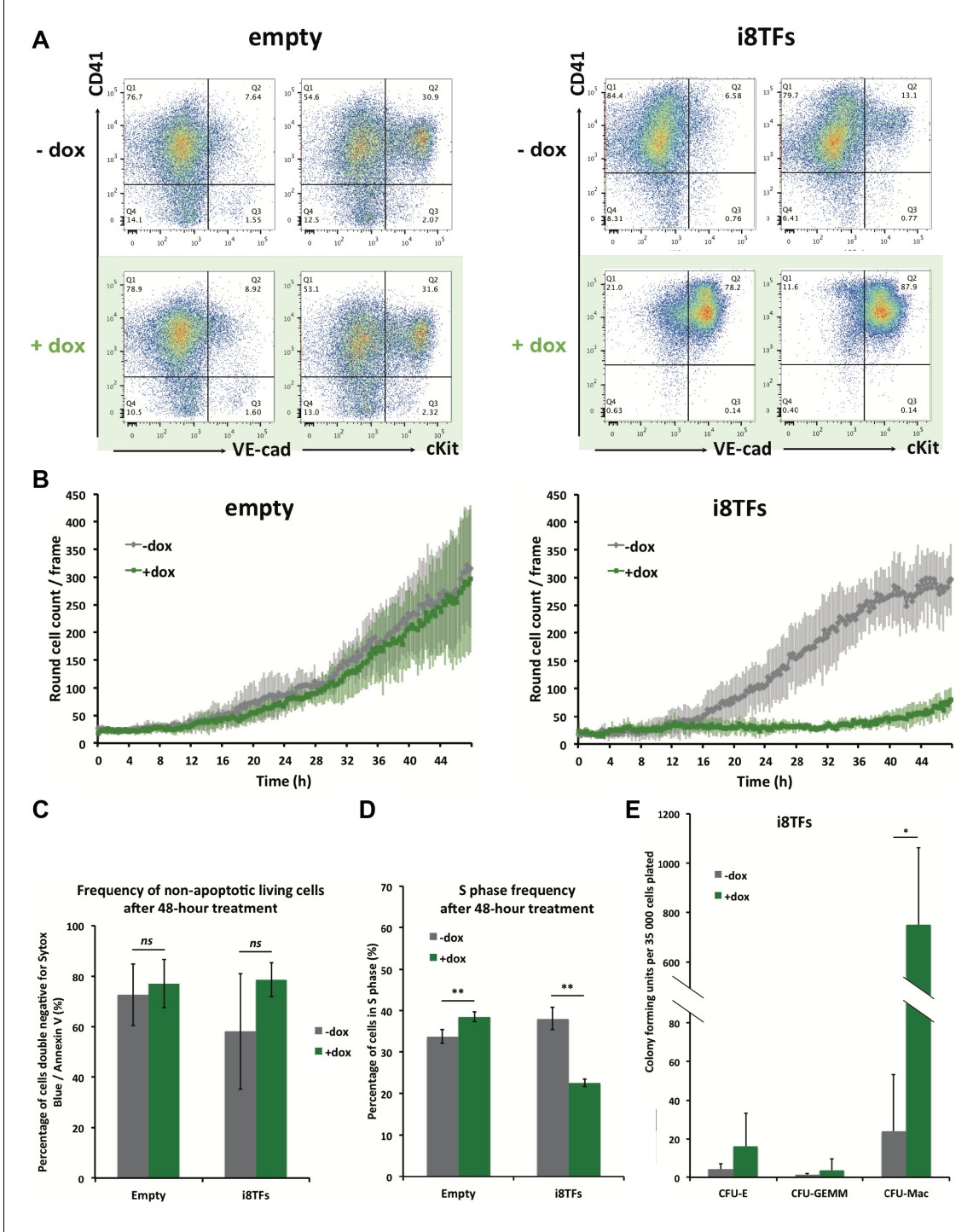

**Figure 3.** Hematopoietic differentiation of the i8TFs ESC line highlights the role of the eight transcription factors in the generation of Pre-HSPCs. (**A**) Representative FACS plots of VE-Cad, cKit and CD41 expression after 3 days of BL-CFC culture of the indicated ESC lines (n = 3). (**B**) Graphs showing the average numbers of round cells counted per frame (n = 3) in a 48 hr time course for the indicated ESC lines. Error bars represent standard deviations. (**C**) Bar graph indicating the frequency of non-apoptotic living cells for the indicated ESC lines. Error bars represent standard deviations (n = 3). (**D**) Bar graph displaying the frequency of cycling cells for the indicated ESC lines. Error bars represent standard deviations (n = 3). (**E**) Bar graph displaying the number of hematopoietic colonies after replating cells in a colony-forming-unit assay from the indicated conditions. Error bars represent standard deviation (n = 4). *ns*, non significant. ** p-value<0.01 (paired two-tailed t-test). * p-value<0.05 (paired two-tailed t-test). See also *Figure 3— figure supplement 1* to 5, *Supplementary file 1*, *4*, *5* and *11*.

DOI: https://doi.org/10.7554/eLife.29312.010

*Figure 3 continued*

The following figure supplements are available for figure 3:

**Figure supplement 1.** Microscopy pictures of the i8TFs BL-CFC culture after one, two and three days of BL-CFC culture in -dox and +dox conditions.
DOI: https://doi.org/10.7554/eLife.29312.011
**Figure supplement 2.** Comparison of eVSM and Endo populations response to the induction of the 8TFs.
DOI: https://doi.org/10.7554/eLife.29312.012
**Figure supplement 3.** PCA plots of microarrays data.
DOI: https://doi.org/10.7554/eLife.29312.013
**Figure supplement 4.** Analysis of microarrays data.
DOI: https://doi.org/10.7554/eLife.29312.014
**Figure supplement 5.** Analysis of the i1TF ESC lines.
DOI: https://doi.org/10.7554/eLife.29312.015

(named eVSMs) (*Figure 3—figure supplement 2B*). To understand the impact of the expression of the eight TFs on the two main non-hematopoietic populations — the VE-Cad$^+$ CD41$^-$ endothelial population (named Endo) and the eVSM — we isolated Endo and eVSM cells at day 1 of BL-CFC culture (*Figure 3—figure supplement 2A*) and cultured them in hemogenic endothelial (HE) culture conditions with or without dox for 48 hr. Using flow cytometry, we showed that both populations were affected by the dox treatment and became almost entirely VE-Cad$^+$ CD41$^+$ (*Figure 3—figure supplement 2C*). Furthermore, the CFU-assay showed an increased hematopoietic differentiation capacity for the Endo +dox compared to the –dox condition, confirming their successful conversion (*Figure 3—figure supplement 2D*). By contrast, the eVSM cells did not show a comparable effect, and together with the flow cytometry data, this evidence suggests that even though the eVSMs gained the Pre-HSPCs phenotype, they probably remained more immature than the Endo +dox cells (*Figure 3—figure supplement 2D*). To further characterize the effect of the eight TFs on the eVSM and Endo populations, we carried out a microarray analysis on i8TFs eVSM and Endo conditions before (t0) and 24 hr after differentiation with or without dox treatment (+dox and –dox) (*Supplementary file 5*). The Empty ESC line eVSM was used as control. As expected, the dox treatment of these cells had little effect on the transcriptome as shown by PCA (*Figure 3—figure supplement 3A*). By contrast, the expression of the eight TFs changed the Endo and eVSM cells dramatically. Interestingly, the eVSM +dox group clustered closely to the Endo -dox and Endo +dox conditions but far apart from the eVSM -dox group. Differentially expressed gene (DEG) analysis highlighted the genes responsible for these changes (*Figure 3—figure supplement 3B* and *Supplementary file 5*). Gene ontology (GO) analysis of the DEGs showed that overexpression of the eight TFs led to the reduced expression of genes involved in vasculature and heart development, while there was an increase in the expression of genes linked to immune system and vasculature development for eVSM (compatible with the dual endothelial-hematopoietic identity of Pre-HSPCs) and cellular detoxification as well as vesicles for Endo (*Figure 3—figure supplement 4A*). Surprisingly, when comparing these DEGs with the 927 targets of the heptad identified by ChIP-seq, the overlap was only 26 genes (*Figure 3—figure supplement 4B and C*). This would suggest that the DEGs that we observed were not a direct consequence of the complex of eight TFs binding at the regulatory elements of these genes.

To find out whether the striking effect of the eight TFs is really due to the expression of several of these genes and not to the effect of one particular transcription factor, we generated eight inducible ESC lines for each of the single transcription factors (i1TF) (*Figure 3—figure supplement 5A*). Inducible expression of each TF was confirmed using western blot (*Figure 3—figure supplement 5B*). Using the same experimental layout as that used previously for i8TFs and Empty ESC lines, we performed BL-CFC assays and FACS analyses with these eight new ESC lines and studied the effect of dox treatment on each of them. We found that none of the i1TF lines were qualitatively similar to the i8TFs (*Figure 3—figure supplement 5C and D*).

To better distinguish differentiated cell populations, we repeated the BL-CFC culture with –dox/ +dox treatment for the ten cell lines followed by sc-q-RT-PCR of the 95 genes used previously (*Figure 4* and *Supplementary file 6*). In total, 854 cells were processed using sc-q-RT-PCR and passed quality control. Six biologically relevant cell clusters could be identified using hierarchical clustering analysis, and they were assigned names based on the expression of specific marker genes

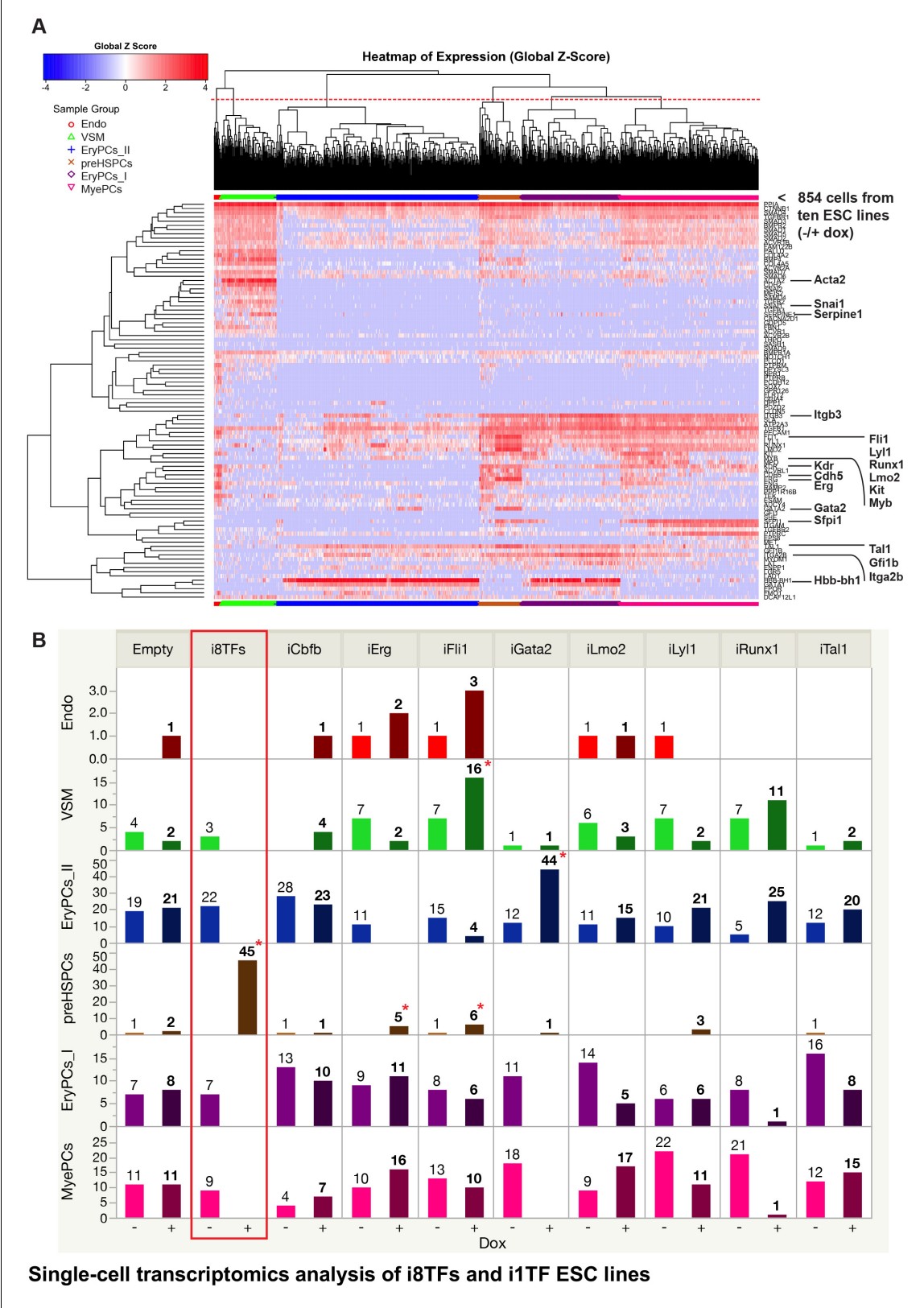

Single-cell transcriptomics analysis of i8TFs and i1TF ESC lines

**Figure 4.** Single-cell transcriptomics analysis of i8TFs and i1TF ESC lines. (A) Hierarchical clustering showing the sc-q-RT-PCR results for 95 genes on the ten inducible ESC lines after three days of BL-CFC culture. The clusters were defined according to the intersection of the red dotted line with the dendrogram in the upper part of the heatmap. (B) Bar graphs displaying the cell number in each of the six clusters defined in (A) for the ten cell lines.
*Figure 4 continued on next page*

*Figure 4 continued*

Stars indicate significant differences for +dox conditions (see *Supplementary file 6* for p-values). See also *Figure 4—figure supplements 1* and *2*, *Supplementary file 1*, *6* and *11*.

DOI: https://doi.org/10.7554/eLife.29312.016

The following figure supplements are available for figure 4:

**Figure supplement 1.** Six scatter plots are displayed corresponding to the six populations defined in *Figure 4*.

DOI: https://doi.org/10.7554/eLife.29312.017

**Figure supplement 2.** Comparison of Empty and i8TFs day 3 BL-CFC cultures by single-cell RNA sequencing.

DOI: https://doi.org/10.7554/eLife.29312.018

(*Figure 4A*). The Endo, VSM and Pre-HSPCs groups were defined as before (*Figure 1*). On the other hand, HSPCs were split into three groups: Erythroid Progenitor Cells I and II (EryPCs_I and EryPCs_II, respectively) and Myeloid Progenitor Cells (MyePCs). EryPCs_I and EryPCs_II were characterized by the expression of erythroid genes such as *Gata1* and *Hbb-bh1*. EryPCs_I appeared to be more immature than EryPCs_II because the cells still express a high level of *Itgb3* and *Pecam1*. MyePCs were characterized by the expression of myeloid markers such as *Sfpi1* (or *Pu.1*) and *Itgam* and by the lack of erythroid genes.

For each cluster, the number of cells per condition was calculated and each +dox condition was statistically compared to all −dox conditions (*Figure 4B* and *Figure 4—figure supplement 1*). This analysis confirmed that following dox addition, the i8TFs cells form a Pre-HSPCs population co-expressing endothelial and hematopoietic genes, as suggested by the flow cytometry results. By contrast, none of the other cell lines presented such a dramatic change. Induction of Gata2 increased the frequency of EryPCs_II population by ~4 fold (*Figure 4B*). On the other hand, overexpression of Erg and Fli1 gave rise to slightly more Pre-HSPCs (*Figure 4B*). The other i1TF ESC lines had negligible changes upon dox treatment (*Figure 4B* and *Figure 4—figure supplement 1*).

Our sc-q-RT-PCR results suggested that the cells were homogeneous after the induction of the eight TFs. To confirm this observation, we carried out single-cell RNA sequencing (sc-RNA-seq) on i8TFs −dox, i8TFs +dox, Empty −dox and Empty +dox cells using the iCELL8 platform (*Figure 4—figure supplement 2A*). Again, PCA showed that i8TFs + dox cells were clearly clustered separately from the other cells (*Figure 4—figure supplement 2B*). On the basis of the PC2 variance, we noticed that there is less variance for the i8TFs +dox cells compared to the other three conditions. Altogether, this would suggest that the i8TFs +dox cells are comparatively more homogeneous than the control cells.

## In vitro generated Pre-HSPCs share similarities with AGM Pre-HSCs type I

To better characterize the i8TFs +dox cells compared to embryonic populations, we examined the expression pattern of CD41, CD43 and CD45 proteins together with VE-cad. It has been shown that these markers could define more precisely several intermediate stages in the EHT process: Pro-HSCs (VE-Cad$^+$ CD41$^+$ CD43$^-$ CD45$^-$), Pre-HSCs type I (VE-Cad$^+$ CD41$^+$ CD43$^+$ CD45$^-$) and Pre-HSCs type II (VE-Cad$^+$ CD41$^+$ CD43$^+$ CD45$^+$) (*Rybtsov et al., 2014*). Our results show that around 99% of VE-cad$^+$ cells in the i8TFs +dox condition have a phenotype similar to Pro-HSCs (*Figure 5A and B*). To verify this finding, we isolated from E10 AGM the Pro-HSCs and Pre-HSCs type I populations and performed sc-q-RT-PCR with our set of 95 genes (*Figure 5—figure supplement 1A and B*). Interestingly, we found that the Pro-HSCs population was heterogeneous. Although all cells express a high level of endothelial genes, only half express blood genes such as *Runx1*, *Gfi1*, *Sfpi1* and *Myb*. By contrast, the Pre-HSCs type I is more homogenous. All cells express endothelial genes but at a lower level compared to Pro-HSCs. Moreover, all cells express hematopoietic genes (*Figure 5—figure supplement 1B*). When we analyzed these cells together with the in vivo data from *Figure 1A* and the cells from i8TFs −dox and +dox cultures, we noticed that about half of Pro-HSCs clustered with Endo, while the other part clustered with the Pre-HSPC (*Figure 5C*). On the other hand, the Pre-HSCs type I clustered with Pre-HSPCs. Surprisingly, in opposition to their Pro-HSCs phenotype, we found that the i8TFs +dox were very close to the Pre-HSCs type I (*Figure 5C*). This was confirmed when the analysis was done without the seven TFs (*Figure 5—figure supplement 1C*) and when the

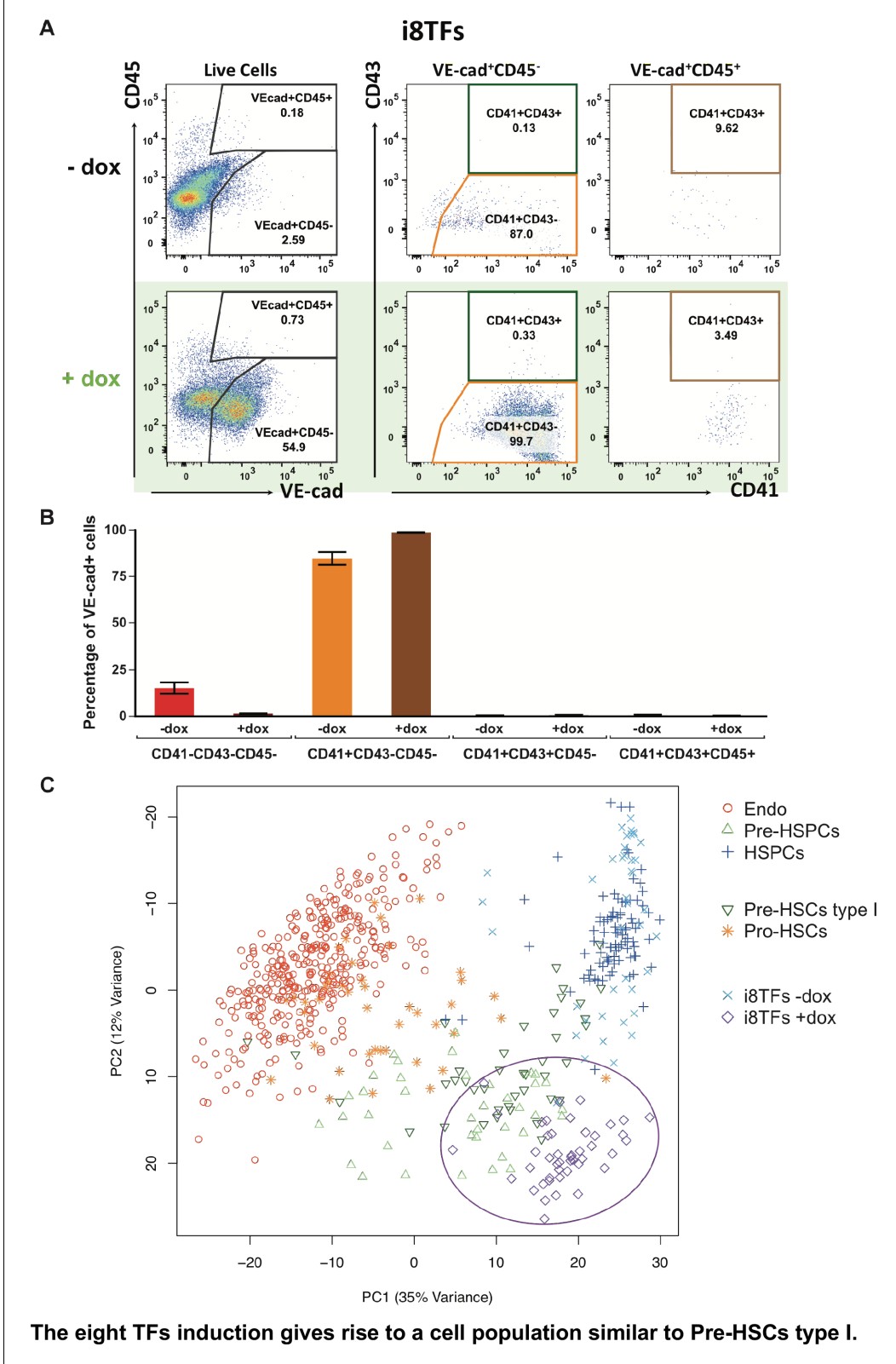

**The eight TFs induction gives rise to a cell population similar to Pre-HSCs type I.**

**Figure 5.** The eight TFs induction gives rise to a cell population similar to Pre-HSCs type I. (**A**) Representative FACS plots of VE-Cad, CD45, CD43 and CD41 expression after 3 days of BL-CFC culture of the i8TFs ESC line in the presence or absence of dox (n = 3). (**B**) Bar graphs showing the average percentage of the four different VE-Cad$^+$ cell populations after three days of culture (n = 3). Error bars represent standard deviations. (**C**) PCA plot showing the sc-q-RT-PCR results for 95 genes combining the cells from the i8TFs cell line after three days BL-CFC culture with the results from cells

*Figure 5 continued on next page*

*Figure 5 continued*

collected from wildtype YS and AGM regions (*Figure 1B*) and from the E10 AGM Pro-HSCs and Pre-HSCs type I. Note that the PC2 axis has been reversed. The ellipse highlights i8TFs +dox. See also *Figure 5—figure supplements 1* and *2*, *Supplementary file 1*, *7*, *8* and *11*.

DOI: https://doi.org/10.7554/eLife.29312.019

The following figure supplements are available for figure 5:

**Figure supplement 1.** Hierarchical clustering analysis of Pro-HSCs and Pre-HSCs type I.

DOI: https://doi.org/10.7554/eLife.29312.020

**Figure supplement 2.** PCA plots comparing the single-cell-q-RT-PCR data of *Figures 1*, *4* and *5*.

DOI: https://doi.org/10.7554/eLife.29312.021

single-cell qRT-PCR data from the 1,660 cells from *Figures 1*, *4* and *5* were combined (*Figure 5— figure supplement 2*).

In conclusion, although the i8TFs +dox cells had a phenotype similar to that of the Pro-HSCs, their transcriptional profile was more similar to Pre-HSCs type I.

## Identification of gene regulatory network from single-cell RNA sequencing data

To determine how the induction of the 8TFs by dox treatment affected GRNs using single-cell-RNA-seq, we decided to focus on the VSM cells because, unlike the Endo population, they are not naturally undergoing EHT (*Figure 3—figure supplement 2C*). Indeed, between Endo +dox and Endo – dox, there was only a two-fold increase in VE-Cad$^+$ CD41$^+$ population. On the other hand, there was a tenfold increase in VE-Cad$^+$ CD41$^+$ population between eVSM +dox and eVSM –dox (*Figure 3— figure supplement 2C*). Moreover, this population shows the largest gene expression contrast in our microarray data upon dox treatment, facilitating the identification of the GRNs controlled by the eight TFs (*Figure 3—figure supplement 3B*). As in the microarray experiments, VE-Cad$^-$ CD41$^-$ cells were sorted at day 1 of BL-CFC culture and cultured in the absence or presence of dox for 24 hr. Subsequently, cells from the –dox and +dox conditions were pooled at a 1:1 ratio and put on a Fluidigm C1 chip. The Fluidigm system allowed a higher sensitivity than the Wafergen system, but it did not allow the processing of several conditions and a large number of cells at the same time. The experimental design whereby cells were mixed and loaded onto one chip minimized the technical variability that would invariably arise from using two separate chips (*Figure 6—figure supplement 1A*). We obtained two biological replicates, and from each chip 96 cells (192 cells in total) were isolated and sequenced together. To determine which cells were exposed to dox we used the expression of the transgene (*Figure 6—figure supplement 1B*). A PCA of the sequencing data showed that cells expressing high levels of the 8TFs transcripts clustered separately from those expressing low levels of these transcripts (*Figure 6—figure supplement 1C*).

To identify regulatory relationships for the eight TFs, we performed a network analysis using the generalized distance correlation measure, dcor statistic (*Székely et al., 2007*), together with its conditional version pdcor (*Székely and Rizzo, 2014*). This measure has the advantage of being able to detect both linear and non-linear relationships between the transcriptome and the seed genes. However, when only the eight TFs were used as seeds, the resulting network was very small with only five of the eight TFs represented (not shown). Thus, we expanded the set of seed genes to include *Gata1*, *Gfi1b*, *Spi1* (or *Pu.1*), *Ldb1* and *Cbfa2t3,* which are known to play important roles in hematopoiesis. Using this extended set of genes, we obtained a network that illustrated how the eight TFs indirectly interacted through a common regulatory pathway (*Ferreira et al., 2005*; *Goardon et al., 2006*; *Imperato et al., 2015*; *Lancrin et al., 2012*; *Mylona et al., 2013*). The directionality of the relationships, positive correlation or negative correlation, was defined using the sign of a two-tailed t-test (gene-pairs with p>0.05 were assigned to 'other'). Finally, interactions between seed genes were identified by conditioning the 'targets' of each seed gene on the expression of each other seed gene. An interaction was inferred if the relationships between both seed genes and a particular 'target' gene increased in strength when conditioning on the other seed gene.

When visualizing the resulting network, we only plotted the seeds that interacted with at least with one other seed (black lines) and the gene targets of at least two seed interactions (*Figure 6A*). Consistent with evidence from the literature, our network reconstruction method reports a tight

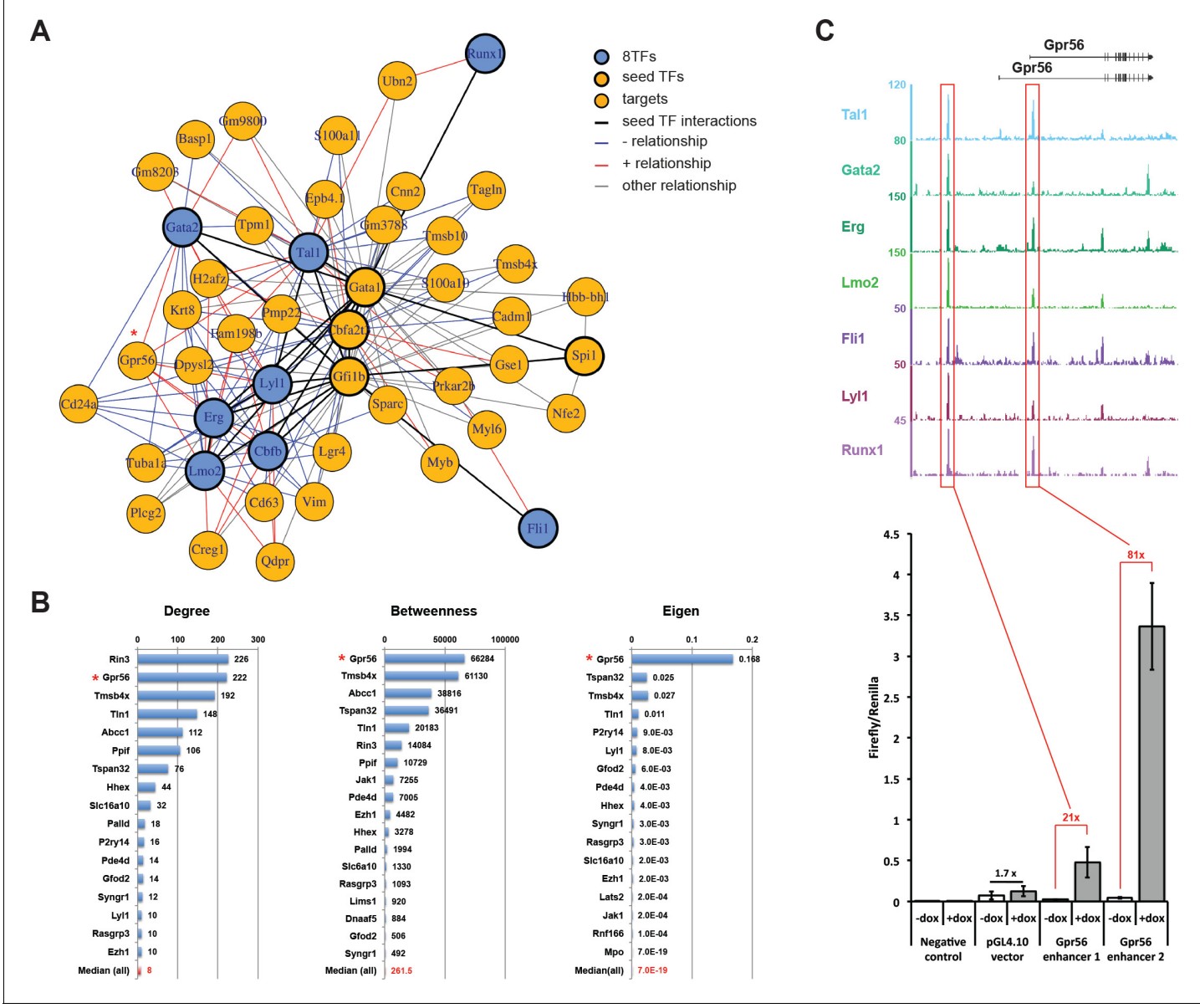

**Figure 6.** Single-cell transcriptomics analysis reveals the transcriptional networks involved in Pre-HSPCs generation. (A) Network built from gene correlations. *Gpr56* is highlighted with a red asterisk. (B) Centrality values for all the genes containing a heptad peak within 1 kb of the TSS that are significantly above the median. *Gpr56* is highlighted with a red asterisk. (C) The upper part shows the ChIP-Seq data for the seven TFs from the *Wilson et al. (2010)* study in the *Gpr56* locus. The y-axis shows reads displayed as density plots, which were generated by *Wilson et al. (2010)* and visualized using the Integrated Genome Browser software. The two putative enhancers are highlighted with red rectangles. The lower part shows the results of the transcriptional reporter assay for two potential heptad enhancers in *Gpr56* locus using i8TFs ESCs treated for 24 hr with doxycycline. Error bars represent standard deviations (n = 3). See also *Figure 6—figure supplement 1* and *Supplementary file 11*.

DOI: https://doi.org/10.7554/eLife.29312.022

The following figure supplement is available for figure 6:

**Figure supplement 1.** Single-cell transcriptome analysis following over-expression of the 8TFs in eVSM cells.
DOI: https://doi.org/10.7554/eLife.29312.023

interaction between the seed factors, apart from *Lbd1*, they are all included in the resulting network. Furthermore, we observe a core network composed of Erg, Lyl1, Lmo2, Tal1, Gfi1b, and Gata1 with Fli1 and Runx1 on opposite ends of the periphery. Several of the genes that were identified as targets of the seed genes were also known from the literature; for example, *Gpr56*, which was upregulated in our microarray experiment after dox treatment (*Figure 3—figure supplement 4C*), and

*Nfe2* (*Wilson et al., 2010*; *Woon Kim et al., 2011*). Moreover, several of the targets had previously been identified as being differentially expressed following dox treatment (*Figure 3—figure supplement 4C*, *Supplementary file 5*).

Encouraged by the results from the dcor analysis, we carried out a genome-wide analysis. Since Dcor is a slow correlation method, it was computationally infeasible to calculate the global network with it. That is why we used Spearman correlation, a very fast correlation method. We combined the two datasets and kept all correlations where the absolute value of the average correlation was greater than 0.25 and where the sign was consistent across Remove Unwanted Variation (RUV) and Size Factors (SF) normalizations. This resulted in a network containing 7,262 genes and 163,474 edges of which 123,394 were positive and 40,080 negative. Importantly, 11 genes out of the 13 seed genes used in the previous analysis where found in this large network: *Cbfa2t3*, *Fli1*, *Gata2*, *Lmo2*, *Erg*, *Tal1*, *Gfi1b*, *Gata1*, *Runx1*, *Spi1* and *Lyl1*. In addition, among the 145 genes containing heptad peaks within 1 kb of their TSS, 26 were found in the network, although this was not significantly more than expected by chance (p>0.1, Fisher-exact test).

To identify the most important genes in the network, we calculated three different centrality measures (degree, betweenness and eigen) (*Figure 6B*). The number of genes among the 11 seed genes with values above the median was significantly higher than that expected by chance for all three measures (degree p-value=0.0005, betweenness p-value=0.006, eigen p-value=0.03). Interestingly, for the 26 heptad targets, only the betweenness presented a higher number of targets above the median than would be expected by chance (p-value=0.04). This highlights the expected central role of the heptad targets by their position 'in between' the main players of the network.

As *Grp56* appeared as a top candidate in our two network analyses, we chose it for further experiments (*Figure 6C*). *Gpr56* has been shown to be expressed in the hematopoietic clusters of the aorta (*Solaimani Kartalaei et al., 2015*) and to be upregulated along the EHT (*Goode et al., 2016*). ChIP-seq data (*Wilson et al., 2010*) show that there are two *Gpr56* regulatory elements bound by seven out of the eight TFs in the HPC7 cell line (*Figure 6C*). We selected these putative enhancers and cloned them in a plasmid upstream of the Firefly luciferase reporter gene. A gene reporter assay in the i8TFs ESC line demonstrated that the induction of the eight TFs led to a strong increase in luciferase gene expression with the two *Grp56* enhancers (*Figure 6C*). In conclusion, we were able to infer GRNs based on sc-RNA-seq data. We identified the key players affected by the induction of the eight TFs, among which *Gpr56* was a major target gene.

## Gene regulatory network analysis reveals contrasting effects of Runx1 and Fli1

The simultaneous expression of the eight factors led to the formation of a population resembling Pre-HSPCs through their specific action on GRNs. It was not clear, however, why the cells were blocked in this normally 'transient' intermediate stage of the EHT. To unravel this, we applied our network analysis to a sc-RNA-seq dataset of endothelial cells and HSPCs, before and after the EHT, respectively (*Pereira et al., 2016*). We generated a network using the same seeds as before (*Figure 6A*), and the most striking feature of the new network is that Fli1 and Runx1 share several targets, but with opposing directions of correlation (*Figure 7A*). A genome-wide network analysis based on Spearman correlations showed a similar pattern in which 210 genes were correlated to both Fli1 and Runx1 but in opposite directions (*Figure 7B* and *Supplementary file 9*). Furthermore, GO analysis revealed that genes whose expression is positively correlated with Fli1 and negatively correlated with Runx1 (yellow rectangle) are enriched for the terms vascular development, angiogenesis and cell migration (*Figure 7—figure supplement 1A*). On the other hand, genes whose expression is positively correlated with that of Runx1 and negatively correlated with that of Fli1 (green rectangle) are associated with the GO term immune system process (*Figure 7—figure supplement 1A*). Hence, Runx1 is more associated with hematopoietic genes whereas Fli1 is more associated with vascular genes. Interestingly, ChIP-seq data from *Wilson et al. (2010)* showed that 77 out of the 210 are direct targets of both Runx1 and Fli1 (*Figure 7—figure supplement 1B* and *Supplementary file 9*). The observed opposite correlation could be due to the fact that most endothelial cells express only Fli1 and not Runx1, while HSPC co-express Runx1 and Fli1. There is indeed a clear shift in the relative expression of Fli1 compared to that of Runx1 between the two cellular stages (*Figure 7—figure supplement 1C*). In conclusion, our network analysis revealed that Fli1 and Runx1 are connected to 210 genes but with opposite correlations, which might be linked to

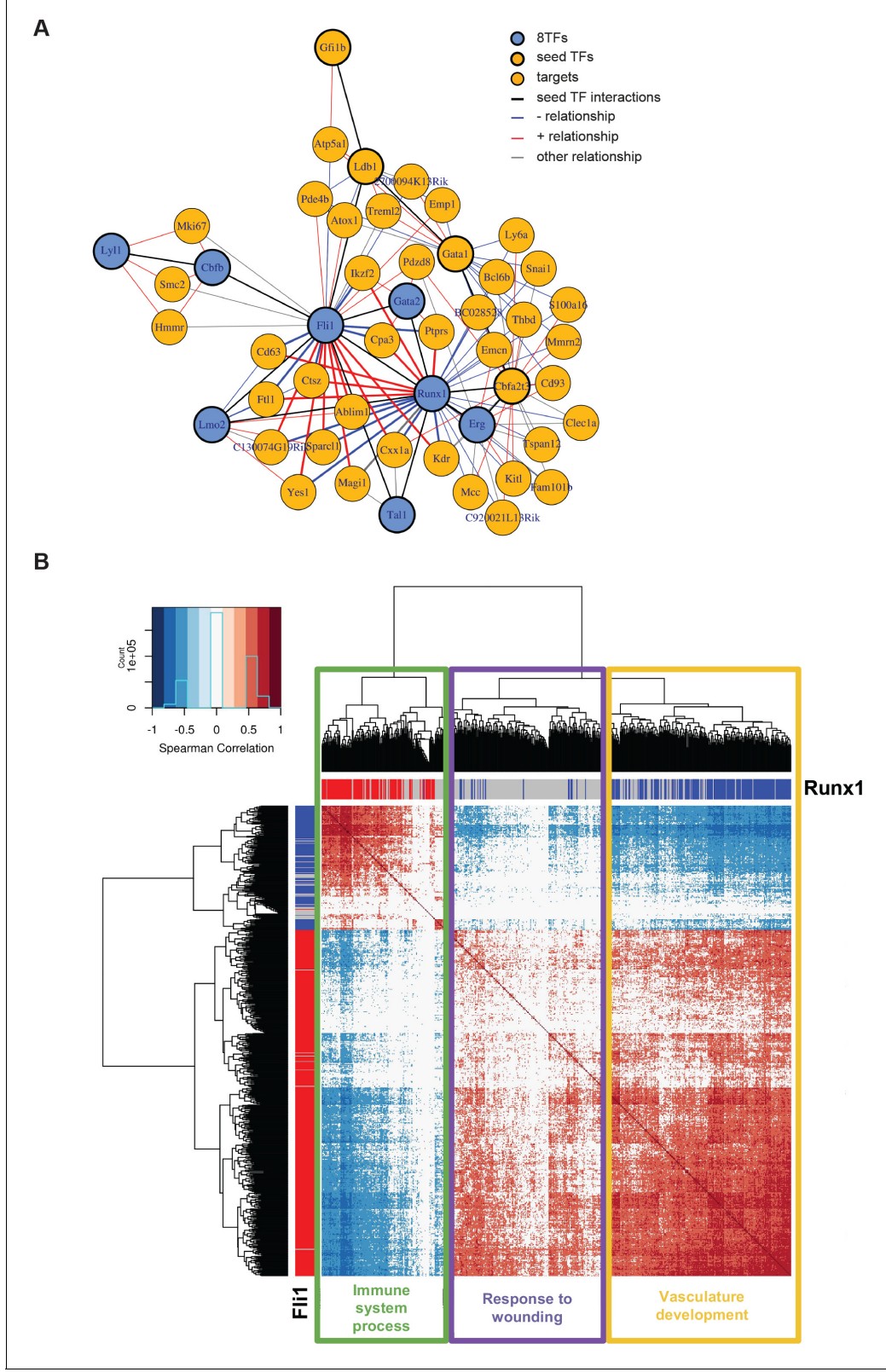

**Figure 7.** Single-cell transcriptome analysis suggests that Runx1 and Fli1 have opposite functions during EHT. (A) Network built from gene correlations found in the Peirera dataset (*Pereira et al., 2016*). (B) Heatmap displaying Spearman correlations between the 210 genes found to be correlated to Runx1 and Fli1. The hierarchical clustering analysis gave us three groups of genes highlighted by three rectangles of different colors. The top GO term for each group of genes is indicated. See also *Figure 7—figure supplement 1* and *Supplementary file 9*.
*Figure 7 continued on next page*

*Figure 7 continued*

DOI: https://doi.org/10.7554/eLife.29312.024

The following figure supplement is available for figure 7:

**Figure supplement 1.** Bioinformatic analysis of the Pereira dataset.

DOI: https://doi.org/10.7554/eLife.29312.025

the contrasting expression patterns of Fli1 and Runx1 and/or to the opposing functional properties of these TFs.

## Gain of function analysis reveals a functional balance between Gata2/ Runx1 and Erg/Fli1 TFs during the endothelial to hematopoietic transition

To find out whether Runx1 and Fli1 have opposite functional effects on the cell fate decisions occurring during EHT, we re-examined our i1TF gain-of-function analysis (*Figure 3—figure supplement 5* and *Figure 4B*). According to these results, we identified two main groups of TFs. The first one includes Fli1 and Erg while the second one includes Runx1 and Gata2. Indeed, in +dox condition, iFli1 and iErg both displayed an increase in Endo population as well as Pre-HSPCs population, although at much lower level than with i8TFs (*Figure 4B*). By contrast, iRunx1 and iGata2 cells showed an increased frequency in the EryPCs_II population and a decrease in the MyePCs population in the +dox condition (*Figure 4B*). This result is consistent with the hypothesis that some of the eight TFs have opposing effects.

We next evaluated the functional consequences of the removal of Runx1/Gata2 or Erg/Fli1 from the 8TFs construct. We hypothesized that removing Runx1 and Gata2 would enhance the endothelial cell fate, whereas removing Erg and Fli1 would increase the hematopoietic cell fate. We consequently generated two additional ESC lines (*Figure 8A*). The first, i6TFs, contains 6TFs that are also found in i8TFs but is without Erg and Fli1. The second, i5TFs, lacks Runx1, Gata2 as well as Cbfb, crucial partner of Runx1. We performed ESC differentiation of these cell lines in the same way as before and analyzed the consequences of the overexpression of these two sets of TFs. Interestingly, the overexpression of the five TFs partly recapitulated the effect of overexpressing the eight TFs and led to a clear increase of the frequency of VE-cad$^+$ CD41$^+$ (*Figure 8B*). Remarkably, the frequency of the VE-Cad$^+$ CD41$^-$ population was significantly increased too, in stark contrast to what we observed with the eight TFs (*Figure 3A* and *Figure 8B*). Consistent with this result, there was a clear decrease of the number of round cells formed during the BL-CFC assay (*Figure 8C*). By contrast, the overexpression of the six TFs only had a minor impact on the expression pattern of VE-cad and CD41 (*Figure 8B*). However, there was a clear increase of round hematopoietic cells as shown by image analysis, suggesting an enhancement of blood cell formation (*Figure 8C*). The observed impact of the induction on the number of round cells was confirmed specifically with endothelial cells isolated at day 1 BL-CFC and cultured for two days in the presence of doxycycline in HE medium (*Figure 8D* and *Figure 8—figure supplement 1A*). In addition, sc-q-RT-PCR performed on the i5TFs and i6TFs confirmed this duality (*Figure 8E* and *Figure 8—figure supplement 1B*). The i5TFs showed a decrease of EryPCs_I hematopoietic progenitors that was associated with an increase in Endo and Pre-HSPC populations, whereas the i6TFs showed an increase in the EryPCs_II population of hematopoietic progenitors (*Figure 8E* and *Supplementary file 10*).

In conclusion, these results suggest that Fli1 and Erg on one side and Runx1 and Gata2 on the other side have opposing effects on the cell-fate decisions in hematopoiesis leading to the formation of Pre-HSPCs.

## Discussion

In this study, we used single-cell transcriptomics to define three main populations in the embryonic vasculature during blood formation: endothelial cells, pre-hematopoietic stem/progenitor cells and hematopoietic stem/progenitor cells. While these results were initially obtained with a single batch of embryos, all key patterns were reproduced in a separate batch in an additional study (Morgan Oatley, unpublished results). Despite slight timing differences, there were strong similarities in gene

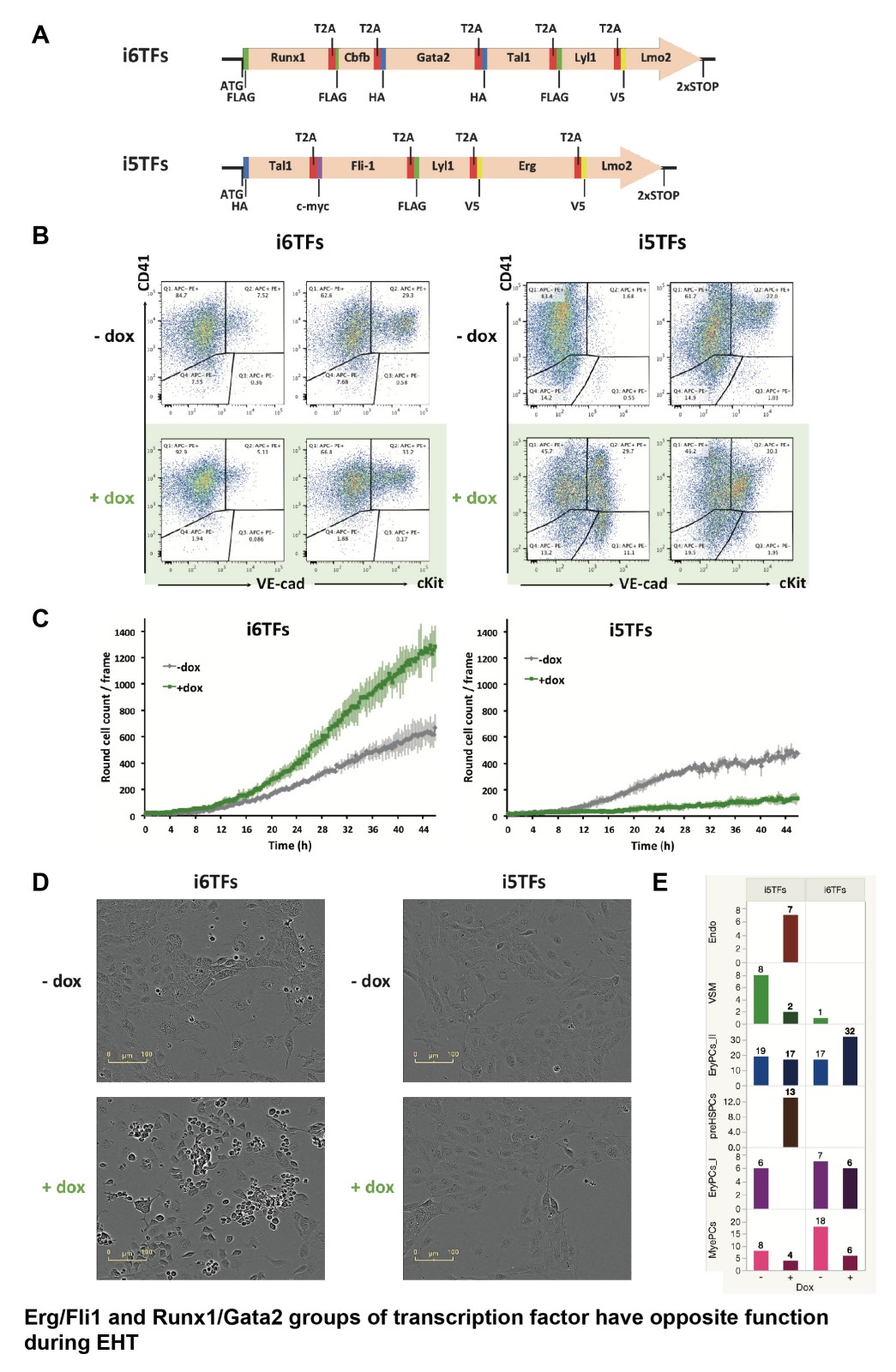

**Erg/Fli1 and Runx1/Gata2 groups of transcription factor have opposite function during EHT**

**Figure 8.** The Erg/Fli1 and Runx1/Gata2 groups of transcription factors have opposite functions during EHT. (**A**) Scheme showing the i5TFs and i6TFs constructs used to generate the two inducible ESC lines missing Runx1, Gata2 and Cbfb, or Fli1 and Erg (respectively). (**B**) Representative FACS plots of VE-Cad, cKit and CD41 expression following three days of BL-CFC culture of the indicated ESC lines (n = 3). (**C**) Graphs showing the average numbers of round cells counted per frame (n = 3) in a 48 hr time course for the indicated ESC lines. Error bars represent standard deviations. (**D**) Representative

*Figure 8 continued on next page*

*Figure 8 continued*

pictures taken two days after HE culture of sorted VE-Cad$^+$ CD41$^-$ (Endo) cells for the indicated cell lines. Round cells correspond to blood cells. The scale bar corresponds to 100 μm. (E) Bar graphs displaying the sc-q-RT-PCR results for the i6TFs and i5TFs ESC lines. See also *Figure 8—figure supplement 1*, *Supplementary files 1, 10* and *11*.

DOI: https://doi.org/10.7554/eLife.29312.026

The following figure supplement is available for figure 8:

**Figure supplement 1.** Study of the role of Erg/Fli1 and Runx1/Gata2 groups of transcription factors in EHT.

DOI: https://doi.org/10.7554/eLife.29312.027

expression between the yolk sac and the AGM EHT, suggesting a similar process in both sites. However, due to the blood circulation, we cannot rule out cellular exchange between these two sites.

Bulk ChIP-seq analyses have suggested that Erg, Fli1, Tal1, Lyl1, Lmo2, Runx1 and Gata2 work together as a heptad (*Wilson et al., 2010*). Here, we show for the first time that the heptad genes are specifically co-expressed at the single-cell level within the Pre-HSPCs, and that this combination was not found in any other population along the transition. This specific co-expression may be conserved in the human species, as it was shown recently in the in vitro human EHT model system that the intermediate population between endothelial and blood cells co-expresses *Runx1*, *Lyl1*, *Tal1*, *Gata2*, *Erg* and *Fli1* (*Lmo2* was not assessed) at the single-cell level (*Guibentif et al., 2017*) similarly to the Pre-HSPC that we described. In contrast to *Guibentif et al. (2017)*, we investigated the functional meaning of the co-expression of these transcription factors. We showed that their enforced expression, together with that of Cbfb, during hematopoietic differentiation gave rise to Pre-HSPCs in culture. This suggests that the co-expression of the heptad genes is necessary for cells to reach the transient Pre-HSPCs stage before pursuing the transition.

These findings are consistent with the hypothesis of the existence of a heptad protein complex. Surprisingly, only ten percent of the differentially expressed genes are heptad targets and only ten percent of the heptad targets are differentially expressed. This suggests that the gene expression changes that we observed were not a direct consequence of the binding of the eight TFs complex at the regulatory elements of these genes. This implies that the formation of Pre-HSPCs is probably due to a combination of individual functions acting on the GRNs rather than to the sole action of an eight-protein complex.

To define these GRNs, we developed a bioinformatics pipeline for sc-RNA-seq data. We characterized the tight relationships involved in the formation of Pre-HSPCs, thereby giving directionality to the protein–DNA interactions previously identified by ChIP-seq (*Goode et al., 2016*; *Wilson et al., 2010*). Network analyses of the Pereira dataset (*Pereira et al., 2016*) together with our in vitro results revealed a new and unexpected opposite relationship between Runx1 and Fli1 functions. In addition, this network suggests that Gata2 and Lmo2 might act as cooperative co-regulators with Runx1 and Fli1, respectively (*Figure 7A*).

On the basis of our data and what is known in the literature, we propose the following model (*Figure 9*). Initially, the endothelial cell fate program is driven by Erg and Fli1, both ETS transcription factors, which is consistent with our functional analyses (*Figure 3—figure supplement 5D* and *Figure 4*) and studies published before (*Asano et al., 2010*; *McLaughlin et al., 2001*). In addition, *Wilson et al. (2010)* showed that more than 90% of the loci targeted by Fli1 are also targeted by Erg in the HPC7 cell line, suggesting functional redundancy. Then, the expression of Runx1 commenses and this — along with the expression of Gata2, which has pro-hematopoietic function — initiates the hematopoietic transcriptional program . Previous data support this functional interaction as *Runx1*$^{+/-}$::*Gata2*$^{+/-}$ compound heterozygous embryos are not viable due to their severe hematopoietic defects, reinforcing the notion that the two proteins are involved in the same biological process (*Wilson et al., 2010*). At this intermediate Pre-HSPCs stage, endothelial genes are progressively downregulated as shown by our data. As Fli1 and Erg gene expression remain stable between the Endo and the Pre-HSPCs stages, this could be explained by a functional switch for these factors in favor of the hematopoietic cell fate and a decrease in the fraction dedicated to the maintenance of the endothelial program at the single-cell level. Specifically, Fli1 may be involved both in endothelial and hematopoietic gene expression depending on the cellular context. Indeed, in presence of Runx1, Fli1 could potentially activate common target genes such as *Gfi1* and *Gfi1b*, which are involved in the repression of endothelial gene expression (*Lancrin et al., 2012*;

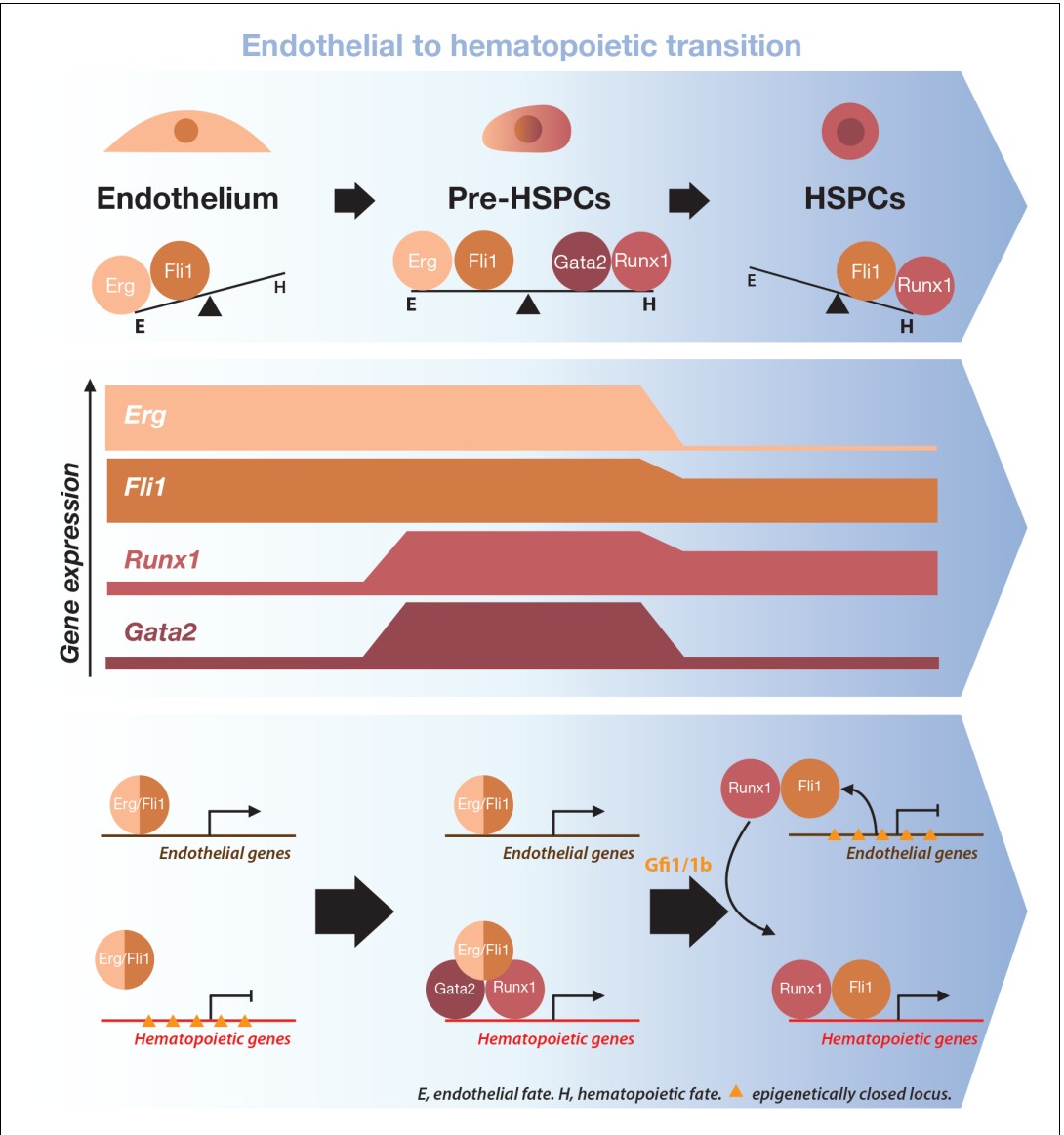

**Figure 9.** Model for the dynamical function of transcription factors during the formation of blood stem and progenitor cells. Proposed model for the dynamical function of transcription factors during the endothelial to hematopoietic transition. The upper blue arrow section shows the balance between transcription factors along the transition. The middle section indicates the gene expression levels for the two sets of transcription factors. In the lower blue arrow section, a possible mechanism for the functional switch of Erg/Fli1 in presence of Runx1/Gata2 is depicted. During the transition, epigenetically closed hematopoietic gene loci are progressively opened and activated in the presence of Runx1/Gata2 together with Fli1/Erg while the endothelial loci are progressively closed through Gfi1 and Gfi1b transcriptional repression. The DNA-binding data combined with single-cell GRN analysis suggests a dual function for Fli1 that is dependent on the cellular context (see main text for full description).
DOI: https://doi.org/10.7554/eLife.29312.028

*Thambyrajah et al., 2016*; *Wilson et al., 2010*). Alternatively (or concomitantly), Runx1 and Fli1 could bind together on endothelial gene loci as shown for *Sox17* (*Lichtinger et al., 2012*) and repress their expression. On the other hand, Fli1 could switch its function upon physical interaction with Runx1, as shown during megakaryocytic differentiation where there is a synergistic transcriptional activation upon Fli1–Runx1 interaction (*Huang et al., 2009*).

At the end of the EHT, in the HSPCs, the endothelial program is completely suppressed along with the expression of Erg. At this development stage, Fli1 is fully dedicated to the hematopoietic

cell fate program and works with Runx1 to carry on blood cell development. This is supported by the observation that almost all the Runx1 target genes specific to the HSPCs are also bound by Fli1 (*Supplementary file 9*).

In conclusion, our work gives an important insight into the switch between endothelial and hematopoietic cell fates that is required for blood cell formation. We have identified the Pre-HSPCs as the pivotal stage in this process, at which a competition between developmental programs takes place at the single-cell level. The expression of the heptad of TFs is directly linked to this transitional stage. The balance of activity of Erg, Fli1, Gata2 and Runx1 appears to have the most important role in the EHT, whereas the Tal1/Lmo2/Lyl1 axis might have an essential role in priming it. Indeed, Tal1 acts as a repressor of alternative cellular programs such as the cardiac lineage and is actively needed to initiate the EHT (*Lancrin et al., 2009*; *Org et al., 2015*). In this context, our single-cell transcriptomics approach allowed us to shed light on an unexpected change in Fli1 activity as a consequence of Runx1 expression. This appears to be a critical process for the proper completion of the EHT. Conditions reinforcing the interaction between Runx1 and Fli1 would probably help to tilt the balance towards the hematopoietic cell fate. Likewise, preventing the Fli1–Runx1 interaction could potentially impair the formation of blood cells from endothelium.

More generally, a dual competing function for a unique transcription factor might be a key process for other developmental transitions such as the epithelial to mesenchymal transition. The bioinformatics approach that we have used in this work could be readily applicable to the study of other developmental processes and could help to address this type of question. Finally, the discrepancy observed between bulk and single-cell analyses in this work could also exist in other studies of TF complexes, which could prompt researchers in different fields to revisit previous assumptions using our approach.

## Materials and methods

### Experimental model and subject details
#### Animals
C57BL/6 N, *Gfi1:GFP* (*Yücel et al., 2004*) and *Gfi1b:GFP* (*Vassen et al., 2007*) mouse lines were used. *Gfi1$^{+/GFP}$* and *Gfi1b$^{+/GFP}$* were crossed with each other to generate *Gfi1$^{+/GFP}$Gfi1b$^{+/GFP}$* animals. These animals were then used for timed mating to generate *Gfi1$^{GFP/GFP}$Gfi1b$^{GFP/GFP}$* double-knockout embryos. Timed mating was set up and the morning of vaginal plug detection was considered day 0.5. Embryos were staged according to morphological properties. All experiments were performed in accordance with the guidelines and regulations defined by the European and Italian legislations (Directives 2010/63/EU and DLGS 26/2014, respectively). All mice were bred and maintained at the EMBL Rome Mouse Facility in accordance with European and Italian legislations (EU Directives 634/2010 and DLGS 26/2014, respectively).

#### Cell lines
##### Generation of inducible ESC lines
All doxycycline-inducible ESC lines were generated using the inducible cassette exchange method described previously (*Iacovino et al., 2011*; *Vargel et al., 2016*). For iCbfb, iRunx1, iLmo2, iGata2 and iFli1 ESC lines, the transcription factor coding sequences were tagged in 5' with an 8His-tag and amplified by PCR from differentiated ESC cDNA. The 8His-tagged coding sequences were then inserted into a pGEM-t-easy before cloning into the p2lox plasmid. For iTal1, iLyl1 and iErg, the 8His-tagged coding sequences were synthetized and cloned into a pUC57 Simple by the GenScript Gene Synthesis Service (http://www.genscript.com/ gene_synthesis.html). The 8His-tagged coding sequences were then cloned from the pUC57 Simple into the p2lox plasmid. For each of the generated p2lox plasmids, a Kozak sequence was then added at the start codon by PCR cloning. The subsequent p2lox plasmids were used to transfect the A2lox.Cre ESC line and to generate the eight single-factor inducible ESC line as previously described (*Vargel et al., 2016*). For the i8TFs ESC line, three individual constructs were synthetized and cloned into a pUC57 plasmid by the GenScript Gene Synthesis Service. Construct one was composed of the coding sequence for V5-Erg-T2A-V5-Lmo2. Construct two was composed of the coding sequence for HA-Tal1-T2A-cmyc-Fli1-T2A-FLAG-Lyl1-T2A. Construct three was composed of the coding sequence for FLAG-Runx1-T2A-FLAG-Cbfb-

T2A-HA-Gata2-T2A. All three constructs were flanked by a start codon and two stop codons, together with the proper restriction sites for cloning into the p2lox plasmid, in order to allow their individual use. They also contain specific restriction sites for their excision, the addition of any of the other constructs, or the removal of any of the factor sequences. Construct one together with the start codon and two stop codons was cloned into the p2lox plasmid. Then, Construct two was added between the start codon and Construct one by classic cloning. Finally, Construct three was added between the start codon and Construct two to obtain the p2lox-8TFs. This plasmid was then transfected into the A2lox.Cre ESC line for the generation of the i8TFs. For the i5TFs, the intermediate p2lox construct containing Construct one and Construct two was used to generate the ESC line. For the i6TFs, V5-Erg and cmyc-Fli1 were successively excised from the p2lox-8TFs by classic cloning to generate the p2lox-6TFs used for the generation of the inducible ESC lines. The empty ESC line was generated using the basic p2lox plasmid and following the same procedure. For each inducible ESC line, a clone was selected based on the induction of the transcript expression measured by qPCR and validated by western blot analysis.

## Identification of cell lines

All cell lines used in this work were mESC lines. The Runx1$^{+/hCD4}$ mESC line was generated in the group of Georges Lacaud by the corresponding author, Christophe Lancrin (*Sroczynska et al., 2009a*). All A2lox.Cre-derived mESC lines were generated from the A2lox.Cre mESC line, which was a gift from Michael Kyba who produced it in his laboratory (*Iacovino et al., 2011*). All A2lox.Cre-derived mESC lines were generated in our laboratory as described in the section 'Generation of inducible ESC lines'. All ESC lines had proper stem cell morphology and were able to give rise to blood and endothelial cells after in vitro differentiation (*Figure 1D*, *Figure 3*, *Figure 3—figure supplements 1* and *5*, *Figure 8* and *Figure 8—figure supplement 1A*). All the inducible mESC lines were verified by western blots and qPCR analysis (*Figure 2*, *Figure 3—figure supplement 5*, *Figure 4* and *Figure 8—figure supplement 1B*).

All cell lines used in this manuscript were tested negative for mycoplasma contamination.

## Maintenance and differentiation of ESC lines

Growth and differentiation of ESCs were performed as previously described (*Sroczynska et al., 2009b*). Briefly, ESCs were maintained on feeders in an ESC culture medium made of KnockOut DMEM (GIBCO) (supplemented with 1% Pen/Strep [GIBCO], 1% L-Glutamine [GIBCO] and 1% NEAA [GIBCO]), 15% FBS (PAA), 0.0024% 1 mg/ml LIF (The Protein Expression and Purification Core Facility, EMBL-Heidelberg) and 0.24% 50 mM 2-mercaptoethanol (GIBCO). For differentiation, ESCs were seeded on gelatin-coated plates (0.1% gelatin [BDH] in PBS for 20 min) for one day in the normal ESC culture medium and then for one day in a medium similar to the ESC medium but containing IMDM (Lonza) instead of the KnockOut DMEM and without NEAA supplement.

To obtain embryoid bodies (EBs), cells were subsequently harvested and cultured in petri dishes at a density of $0.3 \times 10^6$ cells per 10 cm$^2$ dish with an EB medium made of IMDM (Lonza) (supplemented with 1% Pen/Strep [GIBCO] and 1% L-glutamine [GIBCO]), 0.6% transferrin (Roche, Italy), 0.039% MTG (Sigma) and 50 mg/μl ascorbic acid (Sigma). After 3–3.25 days, EBs were harvested and Flk1$^+$ cells were sorted using MACS MicroBead Technology (Miltenyi Biotec). For hemangioblast differentiation, Flk1$^+$ cells were then cultured on gelatin-coated plates at a density of $0.01 \times 10^6$ cells per cm$^2$ in a medium made of IMDM (Lonza), 1% Pen/Strep (GIBCO), 1% L-glutamine (GIBCO), 10% FBS (PAA), 0.6% transferrin (Roche), 0.039% MTG (Sigma), 0.5% ascorbic acid (Sigma), 15% D4T supernatant (EMBL-Monterotondo), 0.05% VEGF (10 μg/ml) (R and D) and 0.1% IL-6 (10 μg/ml) (R and D). During the hemangioblast differentiation, the cell population could be sorted and further cultured in hemogenic endothelium culture conditions. For this purpose, sorted cells were seeded on gelatin-coated plates at a density of $0.015 \times 10^6$ cells per cm$^2$ in a specific medium made of IMDM (Lonza), 1% Pen/Strep (GIBCO), 1% L-glutamine (GIBCO), 10% FBS (PAA), 0.6% transferrin (Roche), 0.039% MTG (Sigma), 0.5% ascorbic acid (Sigma), 0.0024% LIF (EMBL-Heidelberg), 0.25% SCF (20 μg/ml) (R and D), 0.1% oncostatin M (10 μg/ml) (R and D) and 0.01% bFGF/FGF2 (10 μg/ml) (R and D). To pursue the differentiation with a hematopoietic progenitor assay, cells were collected three days after Flk1 sort from the hemangioblast (or the hemogenic endothelium) culture and plated at a density of $3.3 \times 10^3$ cells per cm$^2$ in a medium made of IMDM (Lonza), 1% Pen/Strep (GIBCO), 1% L-glutamine (GIBCO), 55% of 0.5 g/ml methylcellulose (VWR), 15% PDS (Antec), 10%

PFMH-II (GIBCO), 0.6% transferrin (Roche), 0.039% MTG (Sigma), 0.5% ascorbic acid (Sigma), 0.5% SCF (20 µg/ml) (R and D), 0.1% IL-3 (25 µg/ml) (R and D), 0.1% GM-CSF (25 µg/ml) (R and D), 0.04% IL-11 (12.5 µg/ml) (R and D), 0.2% EPO (10 µg/ml) (R and D), 0.2% IL-6 (5 µg/ml) (R and D), 0.2% TPO (12.5 µg/ml) (R and D) and 0.05% MCSF (10 µg/ml) (R and D). Colonies were counted for three replicates after 10–12 days.

For doxycycline treatment of the inducible ESC line, doxycycline (Sigma) was added directly to the culture medium at a final concentration of 1 µg/ml for 24 or 48 hr.

## Method details

### Embryo dissection

Yolk sac and AGM dissections were performed as described before (*Bertrand et al., 2005*) and *Robin and Dzierzak, 2005*). Briefly, pregnant mice were killed by cervical dislocation between E9.0 and E11 of gestation. Uterine horns were collected; the maternal tissues were removed as well as the placenta to isolate the embryos. The yolk sac was torn gently and separated from the embryo proper by tearing off the umbilical and vitelline arteries. Then, the somite pairs of the embryos were counted to determine their developmental stage. To isolate the AGM, the head, tail, limb buds, ventral organs and somites were removed. Yolk sac and AGM samples from the same embryonic development stage were pooled together in the same tube. To generate single-cell suspension from the isolated tissues, AGM and yolk sac samples were subjected to collagenase (Sigma) treatment for 30 min. at 37°C. AGM and yolk sac cell suspensions were then used for flow cytometry.

### Western blot analyses

ESCs were cultured for two passages on gelatin to remove MEFs before being treated with 1 µg/ml of doxycycline (Sigma) for 24 hr. For whole protein extracts, cells were harvested using TrypLE express (GIBCO), washed with PBS and lysed in RIPA buffer (Thermo Scientific) for 30 min at 4°C. Samples were then centrifuged for 5 min at high speed and pellets were discarded. Nuclear extracts were prepared using the Nuclear Extract kit (ActiveMotif) following the manufacturer's instructions. Protein concentration was measured using the Pierce BCA Protein Assay Kit (Thermo Scientific) and equal amounts of –dox and +dox protein samples were loaded onto NuPAGE Novex 10% or 12% Bis-Tris Protein gels (Life Technologies). Migration and wet transfer on a PROTRAN nitrocellulose membrane (PerkinElmer) were performed using the XCell SureLock Mini-Cell Electrophoresis System from Invitrogen. Membranes were stained with Ponceau S solution (Sigma) to control the transfer efficiency and protein amounts. The membranes were then washed with TBST (TBS [20 mM Tris-Base, 154 mM NaCl, pH 7.5], 0.1% Tween 20) and blocked with TBST +5% milk for 45 min at RT. Membranes were then incubated ON at 4°C with primary antibodies diluted in TBST +1% milk as follows: 1:2000 anti-FLAG (Sigma), 1:1000 anti-HA (Sigma), 1:2000 anti-V5 (Life Technologies), 1:5000 anti-Runx1 (Abcam), 1:5000 anti-Cbfb (Abcam), 1:2000 anti-Gata2 (Abcam), 1:2000 anti-Tal1 (Abcam), 1:2500 anti-Fli1 (Abcam), 1:5000 anti-Lyl1 (Abcam), 1:5000 anti-Erg (Abcam), 1:1000 anti-Lmo2 (Abcam), 1:2000 anti-His tag (Abcam), and 1:10000 anti-alpha-tubulin (Sigma) (the last being used as loading control). Following primary antibody incubation, the membranes were washed three times with TBST and incubated for 45 min with HRP-conjugated secondary antibodies diluted in TBST +1% nilk as follows: 1:10000 anti-Mouse HRP (GE Healthcare Life Sciences), 1:10000 anti-Rabbit HRP (Jackson) and 1:10000 anti-Rat HRP (GE Healthcare Life Sciences). The membranes were developed using the Amersham ECL Prime Western Blotting Detection Reagent (GE Healthcare Life Sciences).

### Time-lapse photography

Cell differentiation cultures were imaged from the time of doxycycline addition. Phase-contrast time-lapse images were taken with the IncuCyte HD (Essen Biosciences) inside the incubator, every 15 min, three areas per well. Time-lapse videos were generated with the Fiji software (http://fiji.sc/Fiji) (*Schindelin et al., 2012*) using 10 frames per second. Images were analyzed by CellProfiler software (http://www.cellprofiler.org) (*Kamentsky et al., 2011*) to quantify the number of round cells with a customized pipeline written by Christian Tischer from the EMBL Heidelberg Advanced Light Microscopy Facility. For each time point, the average and standard deviation of counts from all three spots were calculated to make graphs with Microsoft Office Excel software. A detailed explanation on how

time-lapse photography and image analysis were performed can be found at Bio-protocol (*Bergiers et al., 2019*)

## Flow cytometry and cell sorting

Staining was done as described previously (*Sroczynska et al., 2009b*) and analyses were performed with a FACSCanto (Becton Dickinson) or a FACSAria (Becton Dickinson). Cell sorts were performed with a FACSAria (Becton Dickinson) or by using magnetic sorting (MACS MicroBead Technology, Miltenyi Biotec) and anti-APC MicroBeads (Miltenyi Biotech). The monoclonal rat anti-mouse antibodies used were anti-Mouse CD309 (FLK1) APC (Avas12a1, eBioscience), anti-Mouse CD41 PE (MWReg30, eBioscience), anti-Mouse cKit APC (2B8, eBioscience), and anti-Mouse CD144 (VE-Cadherin) eFluor 660 (eBioBV13, eBioscience). FACS data were analyzed using the FlowJo software (Tree Star, Inc.).

## Cell-cycle assay

Cell-cycle assays were performed using the Click-iT Plus EdU Alexa Fluor 488 Flow Cytometry Assay kit from Molecular Probes following the manufacturer instructions. Briefly, cells from hemangioblast, or hemogenic endothelium, culture were treated with 10 µM of EdU for 1 hr. The cells were then harvested and stained with Ghost Dye Red 780 (Tonbo Biosciences) for live/dead staining before staining with cell surface marker antibodies (anti-Mouse CD41 PE [MWReg30, eBioscience] and anti-Mouse CD144 [VE-Cadherin] eFluor 660 [eBioBV13, eBioscience]). Stained cells were fixed and permeabilized, and the Click-IT reaction was performed following the manufacturer instructions. Finally, cells were stained with Hoechst for DNA labeling and analyzed with the FACSAria (Becton Dickinson). FACS data were analyzed using the FlowJo software (Tree Star, Inc.).

## Apoptosis assay

Cells from hemangioblast culture were harvested and stained with anti-Mouse CD41 PE (MWReg30, eBioscience), anti-Mouse CD144 (VE-Cadherin) eFluor 660 (eBioBV13, eBioscience) and AnnexinV (eBioscience). Finally, cells were stained with Sytox Blue for DNA labeling and analyzed with the FACSAria (Becton Dickinson). FACS data were analyzed using the FlowJo software (Tree Star, Inc.).

## Luciferase assay

Sequences from the two heptad peaks found in the Gpr56 locus (*Wilson et al., 2010*) were synthesized and cloned by the GenScript Gene Synthesis Service into a pGL4.10[luc2] reporter plasmid (Promega), with additional 50-bp sequences upstream and downstream to ensure proper peak coverage. The pGL4.74[hRluc/TK] Renilla reporter plasmid that was used to control for transfection efficiency was obtained from Promega. ESCs were plated on gelatin-coated plates (0.1% gelatin [BDH] in PBS for 20 min) and passaged twice to remove the feeders. Cells were then seeded in gelatin-coated 96-well plates at a density of 5,000 cells per well. Next day, the medium was changed and the cells were transfected with the Firefly luciferase reporter plasmids together with the Renilla luciferase reporter plasmid using the NanoJuice Transfection kit (Merck Millipore). After 24 hr, cells were treated with doxycycline (Sigma) for 24 hr and luciferase assay was performed using the Dual-Luciferase Reporter Assay System (Promega) and the POLARstar Omega device (BMG Labtech) following manufacturer's instructions.

## mRNA microarray

Cells were collected and washed once with PBS. Total RNA was extracted with RNAeasy Plus Micro kit (Qiagen). Samples were then tested for quality with BioAnalyzer (Agilent) and their concentrations measured using Qubit (Life Technologies). They were hybridized on an Affymetrix GeneChip Mouse Gene 2.0 ST Array.

## Single-cell quantitative RT-PCR

A maximum of 1 day prior to the experiment, 96-well plates (BioRad) were filled with 5 µl of 2x reaction mix from the CellsDirect One-Step qRT-PCR kit (Invitrogen). On the day of the experiment, cells were harvested and stained for FACS analysis. Cells from Anti-Mouse CD41 PE (MWReg30, eBioscience)/Anti Mouse CD144 (VE-cadherin) eFluor 660 (eBioBV13, eBioscience) staining were single-

cell sorted in the pre-filled 96-well plates (BioRad) using the FACSAria system (Becton Dickinson). The plates were snap-frozen on dry ice immediately after sorting and stored at −80°C for a maximum of 1 week. To perform RT/Specific Target Amplification (RT/STA), 4 µl of a reaction mix — composed of 2.8 µl of Resuspension buffer, 0.2 µl of SuperScript III and Platinum Taq polymerase from the CellsDirect One-Step qRT-PCR kit (Invitrogen) and 1 µl of 500 nM outer primer mix (primer sequence list, see table below) — was added to each well. Plates were first incubated at 50°C for 15 min. followed by incubation at 95°C for 2 min. Then, they were incubated at 95°C for 15 s. followed by incubation at 60°C for 4 min. for 20 cycles. To prepare the sample mixes, 1/5x diluted cDNAs were mixed wiyj the loading reagent (Fluidigm) and SsoFastTM EvaGreen Supermix (Bio-Rad). In parallel, individual assay mixes were prepared as 5 µM inner primer mix, with DNA suspension buffer and the assay loading reagent (Fluidigm). After the priming of 96.96 dynamic array IFC (Fluidigm), the sample mixes and assay mixes were put together in the corresponding inlets and loaded into the chip. Finally, the chip was run in the Fluidigm Biomark HD system by using the Biomark Data Collection Software and the GE96 ×96PCR + Meltv2.pcl program.

To control for technical variation in our sc-qRT-PCR runs, we used the universal cDNA reverse transcribed by Random Hexamer: Mouse Normal Tissues (BioChain). It allowed us to detect consistently 93 out of our 95 genes of interest. A sample without DNA template was also run each time to test for unspecific PCR amplification.

The primer list is as follows:

| Gene name | Outer forward | Outer reverse | Inner forward | Inner reverse |
|---|---|---|---|---|
| Acta2 | aggcaccactgaaccctaag | cacagcctgaatagccacat | ccaaccgggagaaaatgac | atggcggggacattgaag |
| Acvr1 | cgagtgctaatgatgatggctttccc | cgtttcacagtggtcctcgttcc | cgattccccgagtgtggaagatga | cgattccccgagtgtggaagatga |
| Acvr1b | atgctgcgccatgaaaacatc | tcgtgatagtcagagacaagccaca | ttggctttattgctgctgaca | tgggtccaggtgccattatc |
| Acvr2a | gctggcaagtctgcaggtga | tcagaaatgcgtccctttgga | acccatgggcaggttggta | tttatagcaccctccaacacctctg |
| Acvr2b | gaagggctgctggctagatga | agttgccttcgcagcagca | tcaattgctacgacaggcagga | aagtacacctgggggttctcctc |
| Acvrl1 | cgaattgcccatcgtgacctcaa | cgtggttgttgccgatatccaggta | cgaagtcgcaatgtgctggtcaa | cgaagtcgcaatgtgctggtcaa |
| Adcy4 | cgatggtggcatcttgttcccta | cgttcagcttgggttccttcagta | cgacattcccacgcctggctgtc | cgacattcccacgcctggctgtc |
| Atp2a3 | cgaccattgtggctgcagtagaa | cgtcccagaattgctgtgaggaa | cgagagggcagggccatctaca | cgagagggcagggccatctaca |
| Bmp4 | cagccgagccaacactgtga | tgggatgctgctgaggttga | agtttccatcacgaagaacatctgg | gaggaaacgaaaagcagagc |
| Bmpr1a | acgttcaccgaaagcccagcta | tcgcggatgctgccatcaaagaa | acgcgcaggacaatagaatgttg | acgcgcaggacaatagaatgttg |
| Bmpr2 | cagacggccgcatggagtat | atgagccagacggcaagagc | tgcttgtgatggagtattatcccaatg | tacccaatcacttgtgtggagactca |
| Cacna2d1 | acgccaatggttttagctggtgaca | cgtgattggccagtgacgttgaa | acgagcaagttcaatggacaaatgtg | acgagcaagttcaatggacaaatgtg |
| Cdh2 | atcaacaatgagactggggacatca | cttccatgtctgtggcttgaa | cactgtggcagctggtctgg | attaacgtatactgttgcactttctctcg |
| Cdh5 | cgacaccatcgccaaaagagagac | cgtcttagcattctggcggttcac | cgaggatttggaatcaaatgcacatcg | cgaggatttggaatcaaatgcacatcg |
| Cldn5 | tggcaggtgactgccttcct | tgcatgtgcccggtactctg | accacaacatcgtgacggcgcaga | accacgcacgacatccacag |
| Col4a2 | cgaccctggaagccctggattta | cgttgtcttcccttaagtcccaaca | cgatggcagggatgcctggt | cgatggcagggatgcctggt |
| Col4a5 | cgaggactctctgtggattggcta | cgtatgaagggagcggaacgaa | cgattcatgatgcatacaagtgcagga | cgattcatgatgcatacaagtgcagga |
| Ctnnb1 | aactcctgcacccaccatcccca | gttcccgcaaaggcgcatga | tggcctctgataaaggcaactg | tgctgggcaaagggcaag |
| Dcaf12l1 | cgaaggcctccatggcaactac | cgtggtggggagacccagaaaa | cgagcaggcctctggagcta | cgagcaggcctctggagcta |
| Dpysl3 | cgaccattgggaaggacaacttcac | cgtcggccacaaactggttttca | cgaccatccctgaaggcaccaat | cgaccatccctgaaggcaccaat |
| Eng | gcctgactttctgggactccagc | tgctgaccacatgggctgtcac | gccaggctgaagacactgacg | tcatgccgcagctggagtag |
| Enpp1 | cgaccatgaaaggaacggcatca | cgtaatttcctggcttcggatgac | cgatcagcggtcccgtgttt | cgatcagcggtcccgtgttt |
| Epo | ggttgtgcagaaggtcccaga | gggacaggccttgccaaact | ctgagtgaaaatattacagtcccaga | tatggcctgttcttccacctcca |
| Epor | tgaagtggacgtgtcggcag | acagcgaaggtgtagcgcgt | caaccgggcaggagggaca | ccgccccgcaggttgctcagaa |
| Eps8 | cgaaggctgccatgcctttcaa | tcgtgctgttcctcgccacaaa | acgctcctaatcaccaagtagataggaattatg | acgctcctaatcaccaagtagataggaattatg |
| Erg | tcccgaagctacgcaaagaa | tttggactgaggggtgaggtg | tacaactaggccagatttaccttatga | tgtggccggtccaggctgat |
| Esam | cgaagtctatgtctgcaaggctcaa | cgtcccaacaaaagtgcccacaa | cgacagagtgggctttgccaagt | cgacagagtgggctttgccaagt |

*Continued on next page*

*Continued*

| Gene name | Outer forward | Outer reverse | Inner forward | Inner reverse |
|---|---|---|---|---|
| Fam122b | cgaagtttctccagctccttccc | cgtaaatgtcttggattgagaacaggac | cgacaaccagaggatttggaaagcaatg | cgacaaccagaggatttggaaagcaatg |
| Fbn1 | acgtggcggggaatgtacaaaca | cgtcagagctgtgtagcagtaacca | cgactgtcagcagctacttctgcaaat | cgactgtcagcagctacttctgcaaat |
| Fli1 | tgctgttgtcgcacctcagtt | ttccttgacattcagtcgtgagga | ctcagggaaagttcactgctggccta | tggtctgtatgggaggttgtg |
| Flrt2 | cgatgccgctctagcttcttcc | cgtagtggagagcagcctaggaa | cgaccggacccggttggatt | cgaccggacccggttggatt |
| Fmo1 | cgaaaccacgtgaattacggtgta | tcgccttgatgctgggcttgata | cgagctccagaagacaggactca | cgagctccagaagacaggactca |
| Gata1 | cctgtgcaatgcctgtggct | tgcctgcccgtttgctgacaa | gtatcacaagatgaatggtcagaacc | cattcgcttcttgggccggatg |
| Gata2 | aagcaaggctcgctcctg | cacaggcattgcacaggtagt | cagaaggccgggagtgtgtc | gcccgtgccatctcgt |
| Gdpd5 | cgaacttccgacaactcccatacc | cgttgccgatgatgaggctgaa | cgatttctcgggtgccttctcca | cgatttctcgggtgccttctcca |
| Gfi1 | cgagagatgtgcggcaagacc | cgtagcgtggatgacctcttgaa | cgagtgagcctggagcaacacaa | cgagtgagcctggagcaacacaa |
| Gfi1b | cgatggacacttaccactgtgtca | cgtaggttttgccacagacatcac | cgaagtgcaacaaggtgttctcc | cgaagtgcaacaaggtgttctcc |
| Gpr126 | cgactgtgcagccacttcactca | cgtggcagatattccgcacccaata | cgatggagttctgatggatcttcc | cgatggagttctgatggatcttcc |
| Gria4 | cgacgcccatggtgacgaaacta | cgtgccattaccaagacaccatcgta | acgatggatcgctggaagaaactaga | acgatggatcgctggaagaaactaga |
| Hbb-bh1 | gagctgcactgtgacaagcttca | ggggtgaattccttggcaaaa | tggatcctgagaacttcaagc | gagtagaaaggacaatcaccaaca |
| Itga2b | ttccaaccagcgcttcacct | tgctcggatccccatcaaac | cgacaacagcaacccagtgttt | gcccacggctaccgaatatc |
| Itgam | agcaggggtcattcgctacg | cagctggcttagatgcgatgg | attggggtgggaaatgccttc | gtcgagctctctgcgggact |
| Itgb3 | tcctccagctcattgttgatgc | aggcaggtggcattgaagga | acgggaaaatccgctctaaa | agtgacagttcttccggcaggt |
| Kdr | tgtgggggcttgatttcacctg | tcgccacagtcccaggaaag | cactctccaccttcaaagtctcatca | tttcacatcccggtttacaatcttc |
| Kit | ctggctctggacctggatga | cctggctgccaaatctctgtg | tgctgagcttctcctaccaggtg | atacaattcttggaggcgaggaa |
| Lad1 | cgacctctttgagaaggagctgtca | cgtcctgggtcttgctgatcca | cgaggccagaaccgcacagaac | cgaggccagaaccgcacagaac |
| Lat | cgatgctgcctgacagtagtcc | cgttcactctcaggaacattcacgta | cgactgccgtccctgttgtct | cgactgccgtccctgttgtct |
| Lgr5 | cgagcaacaacatcaggtcaatacc | cgtcaggcaaatgctgaaaagca | cgaggagcgagcgttcgtagg | cgaggagcgagcgttcgtagg |
| Lmo2 | tcggccatcgaaaggaagag | gcggtcccctatgttctgct | ctggacccgtctgaggaacc | gcagccaccacatgtcagca |
| Lyl1 | gctgaagcgcagaccaagccat | gctcacggctgttggtgaacact | gtgagctggacttggctgacg | ccgagccaccttctgggggttg |
| Meis2 | cgacccgtacccttcagaagaacagaa | cgttggtcaatcatggctgcacta | cgagaaacagttagcgcaagacac | cgagaaacagttagcgcaagacac |
| Met | cgagatcattggtgcggtctcaa | tcgactcttgcgtcatagcgaac | cgagtagttttgttattatccgggctctt | cgagtagttttgttattatccgggctctt |
| Mpo | tggccctagacctgctgaagag | ttgacacggacaacagattcagc | aagctgcagccctgtgg | gagcaggtgtcaacacatctgtaa |
| Myb | cgagtggcagaaagtgctgaacc | cgttgcttggcaataacagaccaac | cgacatcaaaggtccctggaccaaa | cgacatcaaaggtccctggaccaaa |
| Myom1 | cgattccgtgtacgtgctgtcaa | cgttccgtcatcatccacactcac | acgccaggcaggcgttggaaag | acgccaggcaggcgttggaaag |
| Notch1 | cgaccaaccctgtcaacggcaaa | cgtatttgcctgcgtgctcacaa | cgatgcccctcggggtaca | cgatgcccctcggggtaca |
| Npr1 | cgacagatttgtgggagcttgtacc | cgtcgaaacatccagtccagggta | cgagaccctcccaacatctgtatc | cgagaccctcccaacatctgtatc |
| Palld | cgagcttcgcttcaaggaggac | cgttctggctcctggatgttgaa | cgacttctgaacaatggccaacc | cgacttctgaacaatggccaacc |
| Pcdh12 | cgatggctgcttttgcggaac | tcgtggtttggtttgggctggaa | cgaggaacccggtggagga | cgaggaacccggtggagga |
| Pdzd2 | cgacaacttggaaagcccccaaac | cgtgtccccatttcgtaccatca | cgaagggcaacagtaaaatgaaactcaag | cgaagggcaacagtaaaatgaaactcaag |
| Pecam1 | tgcggtggttgtcattggag | ctggacatctccacgggttt | gtcatcgccaccttaatagttgcag | tgtttggccttggctttcctc |
| Plcd1 | cgatggcttccagtcctagca | cgtcccacgttatggcggacaa | cgatctgggcaggcattctatgagatgg | cgatctgggcaggcattctatgagatgg |
| Ppia | cgaccgactgtggacagctctaa | cgtagtgagagcagagattacaggac | cgatttcttttgacttgcgggcatt | cgatttcttttgacttgcgggcatt |
| Ppp1r16b | cgagcgtgtggatgtgaaggac | cgtgatgtcctggcactgagacta | cgaatggctgggagcctct | cgaatggctgggagcctct |
| Ptprb | acgccaagagcggcaattatgca | cgttgcacccaggacacctttaa | cgaccactccttcaccgaggaa | cgaccactccttcaccgaggaa |
| Ptprc | ggcttcaaggaacccaggaaata | tgacaataactgtggccttttgctc | attgctgcacaagggccccgggatg | cagatcatcctccagaagtcatcaa |
| Ptprm | cgacagacctcctccaacacatca | tcgtctcgtctttcttagcagagtcc | cgatcagatgaagtgcgctgag | cgatcagatgaagtgcgctgag |
| Ramp2 | tcccactgaggacagccttg | tccttgacagagtccatgcaa | tcaaaagggaagatggaagactacga | tcttgtactcataccagcaaggtaggaca |
| Runx1 | cgaactactcggcagaactgagaa | cgtacggtgatggtcagagtgaa | cgaatgctaccgcggccatg | cgaatgctaccgcggccatg |
| Samd4 | cgaacagctccgtccagaagac | cgtactccagcctattgttgatgtca | acgtcgctgcccgtgcata | acgtcgctgcccgtgcata |

*Continued on next page*

*Continued*

| Gene name | Outer forward | Outer reverse | Inner forward | Inner reverse |
|---|---|---|---|---|
| Sash1 | cgattgatctcactgaggagcccta | tcgccatgttggtggcaacatcc | cgactgataagcatggccgttgt | cgactgataagcatggccgttgt |
| Serpine1 | aaaacccggcggcagatcca | cttgttccacggccccatga | gatgctatgggattcaaagtcaa | ccttggagagctggcggagggcatga |
| Sfpi1 | gtgggcagcgatggagaaag | tgcagctctgtgaagtggttctc | atagcgatcactactgggatttctcc | gggaagttctcaaactcgttgttg |
| She | cgaacatggaaccgtacgatgca | tcgtcacagtctcccctggttca | acgtaacagaaatcagacgccgtggtt | acgtaacagaaatcagacgccgtggtt |
| Sla | cgacgaatcttccgtcttcccaac | tcgggtgagcacacagcatagac | cgaactggtactacatctcaccaagg | cgaactggtactacatctcaccaagg |
| Smad1 | tctcagcccatggacacgaa | caccagtgttttggttcctcgt | atgatggcgcctccactgc | gcaactgcctgaacatctcctc |
| Smad2 | tgctctccaacgttaaccgaaa | tcagcaaacacttccccacct | gccactgtagaaatgacaagaagaca | tgtaatacaagcgcactcccttc |
| Smad3 | ccaatgtcaaccggaatgcag | tgaggcactccgcaaagacc | cgtggaacttacaaggcgaca | cccctccgatgtagtagagc |
| Smad4 | ttgcctcaccaccaaaacg | tggaatgcaagctcattgtga | ccatcttcagcaccacccgccta | tggccagtaatgtccaggatg |
| Smad5 | aaccatggattcgaggctgtg | tgacgtcctgtcggtggtactc | tgagctcaccaagatgtgtacc | gctccccagcccttgacaaa |
| Smad6 | ttctcggctgtctcctcctgac | ttcacccggagcagtgatga | gtacaagccactggatctgtccgatt | ggagttggtggcctcggttt |
| Smad7 | ggaagatcaaccccgagctg | tgagaaaatccattgggtatctgga | tgtgctgcaaccccccatcac | aaggaggaggggggagactcta |
| Smad9 | gcacgattcggatgagctttg | tgcagcggtccatgaagatg | gaagggctggggagcagagt | tctcgatccagcagggggtgct |
| Snai1 | cgatctgcacgacctgtggaaa | cgtgagcggtcagcaaaagca | cgactctaggccctggctgctt | cgactctaggccctggctgctt |
| Snai2 | cgagcacattcgaacccacaca | cgttgcagtgagggcaagagaaa | cgattgccttgtgtctgcaaga | cgattgccttgtgtctgcaaga |
| Sox7 | agaacccggacctgcacaac | ccgctctgcctcatccacat | cggagctcagcaagatgc | ggtctcttctgggacagtgtcagc |
| Tal1 | accggatgccttcccatgtt | gcgccgcactactttggtgt | ccaacaacaaccgggtgaaga | aggaccatcagaaatctccatctca |
| Tek | tccaaaggagaatggctcagg | tccggattgttttggccttc | ttccagaacgtgagagaagaacca | tgttaagggccagagttcctga |
| Tgfb1 | accccccactgatacgcctga | gcagtgagcgctgaatcgaa | tggctgtcttttgacgtcactg | gccctgtattccgtctccttgg |
| Tgfb2 | ggcatgcccatatctatggagttc | cagatcctgggacacacagca | gacactcaacacaccaaagtcctca | gggaagcggaagcttcgggattta |
| Tgfb3 | catgtcacacctttcagcccaat | ctccacggccatggtcatct | gagacatactggaaaatgttcatgaggtg | cattgtccactcctttgaatttga |
| Tgfbr1 | gcagacttgggacttgctgtga | catctagaacttcaggggccatgt | catgattctgccacagatacaa | cctttttagtgcctactctgtggtttgg |
| Tgfbr2 | tggccgctgcatatcgtcct | gcatctttctgggcttccatttcca | tggacgcgcatcgccagca | atccgacttgggaacgtg |
| Thpo | ccgacgtcgacccttttgtct | tgcccctagaatgtcctgtgc | tccctgttctgctgcctgct | tgctctgttccgtctgggtttt |
| Upp1 | cgagaaggaagacgtgctctacca | cgtatgaaggtgttcatccgggaa | cgattcaacctcagcactagcacac | cgattcaacctcagcactagcacac |

## C1 single-cell RNA sequencing

After 6,500 cells in 13 µl of PBS were obtained using FACS enrichment, Fluidigm Suspension Reagent was added in a 3:2 ratio and gently mixed by pipetting. A Fluidigm C1 10–17 uM IFC Chip was primed, 5 µl of cell suspension mix was added to the cell inlet, and the chip was loaded into the C1 instrument. Annotation of cell capture was carried out on a brightfield microscope at 40x magnification and each captured site was noted for having single-cell capture, doublet cell capture or debris. Lysis, reverse transcription, and PCR reagents (Clontech Takara) were then loaded into appropriate inlets as per Fluidigm layout. ERCC spike-in controls (Ambion) were added at a dilution of 1:4000 within the lysis mix to gain limit of detection and normalization in downstream analysis. The IFC was loaded into the C1 instrument overnight using the mRNA-Seq RT and the Amp script, and ~3 µl of cDNA was harvested and diluted in 5 µl of Fluidigm C1 DNA dilution buffer. Size distribution and quantification of individual cDNA samples was obtained using an AATI Fragment Analyzer (AATI) and the cDNA concentration was diluted to 100 pg/µl in 10 mM Tris HCl pH 8.0 for library preparation. Illumina library preparation was carried out using the Nextera XT DNA system (Illumina), but volumes were reduced as per the modified Fluidigm C1 protocol. Tagmentation was performed on all 96 samples using 1.25 µl of cDNA (125 pg total concentration) combined with 2.5 µl of tagmentation buffer (TD) and 1.25 µl of Amplicon Tagment Mix (ATM), and incubated for 10 min at 55 degrees. After incubation, 1.25 µl of Neutrilize Tagment Mix (NT) was added. Illumina PCR barcoded indexes were combined to obtain 96 distinct combinations of i7 and i5 barcodes, and 2.5 µl of combined index was added along with 3.75 µl of Nextera PCR Mix (NPM). Samples were then put through 12 cycles of PCR as per the Illumina protocol. Each column was then

pooled into a 0.2 mL 8-strip PCR tube and Ampure XP purification was performed, after which each pool was combined to obtain a single 96-sample multiplexed pool of barcoded library. The final sequencing library was quantified by Qubit (Life Technologies) and size distribution was measured on an Agilent BioAnalyzer. Samples were sequenced with the Illumina HiSeq 2000.

## Wafergen single-cell RNA sequencing

Cells were stained with the Cell Viability Imaging kit (Molecular Probes), which contains Hoechst 33342 and propidium iodide, and afterwards were counted with the Moxi Z Mini Automated Cell Counter (ORFLO). Stained cell solution was diluted in a mix with diluent and RNase inhibitor (New England Biolabs) to 1 cell/50 nl for dispensing on the ICell8-chip (Wafergen) with the MultiSample NanoDispenser (Wafergen). Positive and negative controls were prepared according to the ICell8 protocol and dispensed with the MSND into the respective nanowells of the chip. All nanowells of the ICell8 chip were imaged with a fluorescence microscope (Olympus). The images were analyzed with the CellSelect software (Wafergen). Alive single cells, which are Hoechst-33342-positive and propidium-iodide-negative, were selected for lysis and reverse transcription inside the ICell8 chip (200 cells i8TFs + dox, 150 cells i8TFs –dox, 150 Empty –dox and 150 Empty +dox). RT reaction mix containing 5X RT buffer, dNTPs, RT e5-oligo (Wafergen), nuclease-free water, Maxima H Minus RT (Thermo Scientific) and Triton X-100 was prepared and dispensed into the previously selected nanowells with single cells inside. The chip was placed inside a modified SmartChip Cycler (Bio-Rad) for the RT reaction (42°C for 90 min, 85°C for 5 min, 4°C forever).

The cDNA of all single cells was collected together and further concentrated with the DNA Clean and Concentrator−5 kit (Zymo Research). The Exonuclease I (New England Biolabs) reaction of the cDNA (37°C for 30 min, 80°C for 20 min, 4°C forever) was performed inside a conventional thermal cycler. Afterwards, the cDNA was amplified with the Advantage 2 PCR Kit (Clontech Takara) containing buffer, dNTPs, Amp Primer (Wafergen), polymerase mix and nuclease-free water (95°C for 1 min, 18 cycles of 95°C for 15 s, 65°C for 30 s and 68°C for 6 min, followed by 72°C for 10 min and 4°C forever). The amplified cDNA was purified with Ampure XP Beads (Beckmann Coulter). The cDNA size distribution was obtained with the High Sensitivity DNA BioAnalyzer (Agilent) and quantification was performed with the Qubit (Life Technologies). Illumina library preparation was carried out by using Nextera XT DNA (Illumina). Tagmentation was performed in tagment DNA buffer, Amplicon Tagment Mix and 1 ng of purified cDNA (55°C for 5 min and 10°C forever), next Neutralize Tagment Buffer was added (room temperature for 5 min). After incubation, the NexteraXT PCR reaction mix was prepared with Nextera PCR Mastermix, i7 Index Primer from the Nextera Index Kit (Illumina), Nextera Primer P5 and Tagmented cDNA-NT buffer mix (72°C for 3 min, 95°C for 30 s, 12 cycles of 95°C for 10 s, 55°C for 30 s and 72°C for 30 s, final 72°C for 5 min and 10°C forever). Ampure XP purification was performed with the finished library. The size distribution was checked on an Agilent BioAnalyzer. Samples were sequenced with the Illumina NextSeq.

## Quantification and statistical analysis

### GO term enrichment analysis

All GO term enrichment analyses were performed using the g:Profiler web server (*Reimand et al., 2016*).

### Flow cytometry analysis

All flow cytometry experiments were independently repeated three times. For *Figure 3C* and *Figure 3D*, paired t-test was performed on the cell frequencies. For *Figure 3—figure supplement 5D*, frequencies for each population were plotted using JMP software (SAS). Error bars correspond to standard deviations.

### Single-cell quantitative RT-PCR data analysis

Single-cell quantitative RT-PCR data were first processed with the Fluidigm Real Time PCR Analysis software (quality threshold, 0.65; Ct threshold, Auto (Global); and baseline correction method, linear [derivative]). Hierarchical clustering and PCAs were performed using the SINGuLAR Analysis toolset (Fluidigm version 3.5) package in R (*R Core Team , 2016*). Using this toolset, the Ct values were converted into log2ex. Ct value is a relative gene expression value in the log2 domain (assuming that

the PCR efficiency is equal to 1). The conversion from Ct to log2ex value is based on the following formulae:

Log2ex = LOD – Ct, if Ct is less than the limit of detection (LOD)

Log2ex = 0, if Ct is equal to or greater than LOD.

Where the default is 24 (the typical Ct value for a single copy of input in the Fluidigm chip). In our case, the Ct value is a relative expression and does not have unit. Hence, following conversion to the Log2ex domain, we labelled the resulting values 'Log2 Gene Expression'.

Hierarchical clustering and PCA analyses were performed with the HC() and PCA() functions of this package. The HC function clusters genes by the Pearson Correlation method, that is, co-profiled genes are clustered together and samples are clustered by normalized Euclidian distance (distance/ number of genes), representing the average fold change. The HC analysis uses the 'complete linkage' method (a bottom-up method) to find similar clusters. The 'global_z_score' display option normalizes the expression value using the global mean and the global standard deviation.

The gene expression average of hematopoietic and endothelial genes is shown for each single cell in *Figure 1—figure supplement 3*. The hematopoietic average is based on the genes *Epo, Epor, Gata1, Gata2, Gfi1, Gfi1b, Hbb-bh1, Itga2b, Itgam, Itgb3, Kit, Lmo2, Lyl1, Mpo, Myb, Ptprc, Runx1, Sfpi1, Sla, Tal1* and *Thpo* and the endothelial average on *Cdh5, Cldn5, Eng, Esam, Fbn1, Gpr126, Kdr, Npr1, Pcdh12, Pecam1, Ptprb, Ptprm, Ramp2, Sox7* and *Tek*.

A prototype was calculated for each cluster group of single cells determined by the hierarchical clustering result shown in *Figure 1—figure supplements 1* and *2*. The prototype is defined by the average of all cells belonging to its cluster. The Pearson correlation coefficient between all pairwise prototypes is visualized in *Figure 1—figure supplement 4A*. The R package *ComplexHeatmap* was used (*Gu et al., 2016*).

A generalized linear model with Quasi-Poisson distribution and logarithm as the link function was applied to model the count table data as shown in *Figure 4B*. Six control groups were defined, one for each cell cluster category such as Endo or VSM. Each control group comprises 10 samples and includes all –dox conditions. For each cluster category and for each of the overexpression cell lines, the count of the +dox condition was compared to the counts in its respective control group. Out of these 60 statistical contrasts (six cluster categories times nine overexpression cell lines plus the cell line carrying an empty vector) five were identified to be significantly different with an adjusted p-value<0.05. The method of Benjamini-Hochberg was used for the multiple testing correction. In addition, the generalized linear model included an offset to model rates instead of counts. The rates account for small differences in the total number of cells measured on each of the processed well plates. All five significantly different conditions are marked with a star in *Figure 4B* (see *Supplementary file 6* for p-values).

## mRNA microarray data analysis

The mRNA microarray data were analyzed using *oligo, phylobase, methods, biobase, stats, ggplot2, gplots, matrixstats, graphics, annotationdbi* and *limma* packages in R (*R Core Team , 2016*).

## C1 single-cell RNAseq data analysis

Each library was sequenced generating 426,065,795 and 440,774,932 paired-end reads (of 125 bp). Reads were assigned to the well with the matching barcode. Up to two mismatches are tolerated within the well barcode if the assignment remained non-ambiguous. STAR, version 2.4.0 (*Dobin et al., 2013*), aligned the reads to the mouse genome build mm10/GRCm38. Duplicate reads were removed using samtools, version 0.1.18 (*Li et al., 2009*). Wells with fewer than 1 million reads were deemed to be empty wells and excluded from further analysis (2 and 1, respectively). Reads with a minimum mapping quality of 30, which mapped to a unique gene, were counted using featureCounts, version 1.4.6 (*Liao et al., 2014*).

Read counts were normalized using RUVg, version 1.0.0 (*Risso et al., 2014*), using ERCC spike-in controls and size-factors (SF) as defined in DESeq (*Anders and Huber, 2010*). Cells with >7% of reads mapped to rRNA (three in each plate) and which were outliers on the basis of the first principal component after size factor normalization were excluded. The remaining cells all contained >7,000 detected genes and were used in all subsequent analyses.

We performed a network analysis using the generalized distance correlation measure, dcor statistic (*Székely et al., 2007*), together with its conditional version, pdcor (*Székely and Rizzo, 2014*). For each chip and each normalization method (four configurations), all significant relationships between the transcriptome and the seed genes were identified using the dcor statistic. The seed genes used for this analysis included the 8 TFs as well as Gata1, Gfi1b, Spi1, Ldb1 and Cbfa2t3. The additional seed genes were selected if they appeared in networks published by at least two different authors (*Beck et al., 2013*; *Bonzanni et al., 2013*; *Moignard et al., 2013*; *Moignard et al., 2015*; *Tanaka et al., 2012*). The directionality of the relationships (positive correlation, negative correlation or 'other') was defined using a two-tailed t-test, direct was based on the sign of the test and 'other' was assigned to all genes with uncorrected p-value>0.05. To ensure robustness, we only included relationships that were identified in at least two of the four configurations, and any relationships with conflicting directionalities were assigned to 'other'. Finally, interactions between seed genes were identified by conditioning the 'targets' of each seed gene on the expression of each other seed gene. An interaction was inferred if the relationships between both seed genes and a particular 'target' gene increased in strength when conditioned on the other seed gene.

For the Spearman network analysis, we combined Exp1 and Exp2 into a single expression matrix for each normalization (SF and RUV). We calculated Spearman correlations using base R-function for each normalization and we averaged the correlations across the two normalizations. We changed all correlations with absolute value <0.25 (i.e <0.25 and >−0.25) to 0. We created a network with weighted edges between genes, correlations of 0 = no edge. We calculated eigen centrality using *eigen_centrality* from the *igraph* R package (*Bonacich, 1987*). We took the absolute value of all edges (turns all relationships to be positive) and calculated betweenness centrality (number of shortest paths passing through a node) using *betweenness* from the *igraph* R package (*Brandes, 2001*). We calculated degree (number of edges attached to a node) using *degree* from the *igraph* R package. Then, we calculated the median value of each measure across all genes in the network. We counted the number of genes in the list that are above the median and we tested the significance of this using a binomial test (by definition 50% of genes are above the median).

## Wafergen single-cell RNAseq data analysis

673 wells on the Wafergen chip passed microscopic inspection. Pooling, library construction and sequencing generated 179 million paired-end sequences using the Illumina Nextera protocol. Note that this shallower sequencing greatly reduces the detection rate of TFs, but the larger number of cells facilitated evaluation of heterogeneity within and between populations. The first read of the pair encodes the well barcode and the UMI (unique molecular identifier) barcode, the second encodes the 3′ end of the transcript. Reads were assigned to the well with the matching barcode. Up to two mismatches are tolerated within the well barcode if the assignment remained non-ambiguous. Cutadapt trimmed poly-A, poly-G and adapter sequences and STAR aligned the reads to the mouse genome build mm10 (*Dobin et al., 2013*; *Martin, 2011*). FeatureCounts determined all reads in a strand-specific way so that they overlap with exactly one gene (*Liao et al., 2014*). Gene expression levels are calculated by counting the number of distinct UMIs of all gene specific reads ('UMIs per gene'), as determined by featureCounts. Only uniquely aligned reads were counted. To further reduce the impact of mapping and sequencing errors on the quantification, an UMI barcode error correction and filtering step was introduced. If two or more of the distinct UMIs per gene share a similar barcode, differing by at most one or two mismatches, then they may originate from the same UMI barcode. The UMI barcode is corrected only if it has less than 10% of reads compared to the most abundant one. This correction procedure starts with the UMI barcode, which is supported by the highest number of gene-specific reads, and which determines all similar barcodes and corrects them if applicable. It restarts with all UMI barcodes that have not been considered in a previous iteration until none is left. In a separate filtering step, any of the distinct UMIs per gene that is only supported by one or two aligned reads is completely omitted from the counting. From the resulting count table, all counts originating from the 23 control wells were removed for the subsequent explorative analysis. All remaining wells correspond to single cells. Additional filtering reduced the sparseness found in this gene per cell matrix of UMI counts. Only cells with 1000 or more genes with at least one UMI count and genes with at least one UMI count in 10 or more cells were selected, leaving a matrix of 478 single cells and 10,238 genes. This matrix was scaled and

normalized with the Bioconductor package *scater* (*McCarthy et al., 2017*). The principal components were calculated for the 400 genes that showed the highest variance after subtracting the mean-variance trend of all genes. The explorative analysis was performed using R (*R Core Team , 2016*).

## Data and software availability

### Software

All software is freely or commercially available and is listed in the Methods description and product reference table.

### Data resources

The accession number for all sc-RNA-seq and microarray data reported in this paper is GEO: GSE96986.

## Product reference table

| Product or resource | Supplier | Reference N° |
|---|---|---|
| Antibodies | | |
| Anti-6X His tag antibody [HIS.H8] | Abcam | ab18184 |
| Anti-alpha-Tubulin-Antibody (mouse) | Sigma | T9026 |
| Anti-APC MicroBeads | Miltenyi Biotec | 130-090-855 |
| Anti-Cbfb Antibody | Abcam | ab133600 |
| Anti-Erg Antibody | Abcam | ab92513 |
| Anti-FLAG M2 antibody | Sigma | F1804 |
| Anti-FLI1 antibody [EPR4646] | Abcam | ab133485 |
| Anti-GATA2 antibody [EPR2822(2)] | Abcam | ab109241 |
| Anti-HA antibody produced in rabbit | Sigma | H6908 |
| Anti-Lmo2 Antibody | Abcam | ab91652 |
| Anti-Lyl1 antibody [KT43] | Abcam | ab53354 |
| Anti-Mouse CD144 (VE-Cadherin) eFluor 660, eBioBV13 | eBioscience | 50-1441-80 |
| Anti-Mouse CD309 (FLK1) APC, Avas12a1 | eBioscience | 17-5821-81 |
| Anti-Mouse CD41 PE, MWReg30 | eBioscience | 12-0411-81 |
| Anti-Mouse cKit APC Antibody | eBioscience | 17-1171-82 |
| Anti-Mouse HRP | GE Healthcare Life Sciences | NA931V |
| Anti-Rat HRP | GE Healthcare Life Sciences | NA935V |
| Anti-Runx1 Antibody | Abcam | ab92336 |
| Anti-Tal1 Antibody | Abcam | ab119754 |
| Anti-V5 Antibody | Life Technologies | 46–0705 |
| Peroxidase AffiniPure Goat Anti-Rabbit IgG (H + L) | Jackson | 111-035-144 |
| Chemicals | | |
| 2-mercaptoethanol | GIBCO | 31350–010 |
| Ascorbic acid | Sigma | A4544 |
| Collagenase Type IA from *Clostridium histolyticum* lyophilized powder (from sterile-filtered solution), 0.5–5.0 FALGPA units/mg solid, cell culture tested | Sigma | C9722-50MG |
| D4T supernatant | EMBL-Rome | N/A |
| Doxycycline | Sigma | D9891 |
| EPO | R and D | 959-ME-010 |
| ERCC RNA Spike-In Mix | Ambion | 4456740 |

*Continued on next page*

*Continued*

| Product or resource | Supplier | Reference N° |
| --- | --- | --- |
| Exonuclease I | New England Biolabs | M0293L |
| Fetal bovine plasma-derived serum platelet poor (PDS) | Antech | N/A |
| Fetal bovine serum (FBS) | PAA | A15-102 |
| Fluidigm C1 DNA Dilution Buffer | Fluidigm | 100–5317 |
| Fluidigm Suspension Reagent | Fluidigm | 100–6201 |
| Gelatin | BDH | 440454B |
| Ghost Dye Red 780 | Tonbo Biosciences | 13–0865 |
| GM-CSF | R and D | 425 ML |
| Human FGF basic | R and D | 233-FB-025 |
| IL-11 | R and D | 418 ML |
| IL-3 | R and D | 403 ML |
| IL-6 | R and D | 406 ML |
| IMDM | Lonza | BE12-726F |
| KnockOut DMEM | GIBCO | 10829–018 |
| L-glutamine | GIBCO | 25030–024 |
| LIF | EMBL Heidelberg | N/A |
| Maxima H Minus Reverse Transcriptase | Thermo Scientific | EP0752 |
| MCSF | R and D | 416 ML-010 |
| MEM Non-Essential Amino Acids Solution | GIBCO | 11140–035 |
| Methylcellulose | VWR | 9004-67-5 |
| Monothioglycerol (MTG) | Sigma | M6145 |
| Oncostatin M | R and D | 495-MO |
| Penicillin-streptomycin | GIBCO | 15140–122 |
| PFMH-II | GIBCO | 12040–077 |
| Pierce RIPA Buffer | Thermo Scientific | 89900 |
| Ponceau S Solution | Sigma | P7170 |
| RNase Inhibitor | New England Biolabs | M0314S |
| SCF | R and D | 455-MC |
| TPO | R and D | 488-TO-005 |
| Transferrin | Roche (Italy) | 10652202001 |
| TrypLE express | GIBCO | 12605–036 |
| VEGF | R and D | 293-VE |
| Commercial assays | | |
| Advantage 2 PCR Kit | Clontech Takara | 639207 |
| Amersham ECL Prime Western Blotting Detection Reagent | GE Healthcare Life Sciences | RPN2232 |
| Annexin V-FITC Apoptosis Detection Kit | eBioscience | 88-8005-72 |
| Cell Viability Imaging kit | Molecular Probes | R37610 |
| CellsDirect One-Step qRT-PCR kit | Invitrogen | 11753100 |
| Click-iT Plus EdU Alexa Fluor 488 Flow Cytometry Assay kit | Molecular Probes | C10633 |
| DNA Clean and Concentrator—5 kit | Zymo Research | D4013 |
| Dual-Luciferase Reporter Assay System | Promega | E1910 |
| ICell8-chip and reagent kit | Wafergen | 430–000233 |
| NanoJuice transfection kit | Merck Millipore | 71902 |

*Continued*

| Product or resource | Supplier | Reference N° |
|---|---|---|
| Nextera Index Kit | Illumina | FC-131–1001 |
| Nextera XT DNA Library Prep kit – 24 samples (Wafergen experiment) | Illumina | FC-131–1024 |
| Nextera XT DNA Library Prep kit – 96 samples (C1 experiment) | Illumina | FC-131–1096 |
| Nuclear Extract kit | ActiveMotif | 40010 |
| Pierce BCA Protein Assay kit | Thermo Scientific | 23225 |
| RNAeasy Plus Micro kit | Qiagen | 74034 |
| SMARTer Ultra Low RNA Kit for the Fluidigm C1 System, 10 IFCs | Clontech Takara | 634833 |
| Deposited data | | |
| Raw and analysed sequencing and microarray data | This paper | GEO: GSE96986 |
| Raw and analysed sequencing data | (*Goode et al., 2016*) | GEO: GSE69101 |
| Raw and analysed sequencing data | (*Pereira et al., 2016*) | GEO: GSE54574 |
| Cell lines | | |
| A2lox.Cre mESC line | (*Iacovino et al., 2011*) | A2lox.Cre |
| A2lox.empty mESC line | This paper | Empty |
| A2lox.i8TFs mESC line | This paper | i8TFs |
| A2lox.iCbfb mESC line | This paper | iCbfb |
| A2lox.iErg mESC line | This paper | iErg |
| A2lox.iFli1 mESC line | This paper | iFli1 |
| A2lox.iGata2 mESC line | This paper | iGata2 |
| A2lox.iLmo2 mESC line | This paper | iLmo2 |
| A2lox.iLyl1 mESC line | This paper | iLyl1 |
| A2lox.iRunx1 mESC line | This paper | iRunx1 |
| A2lox.iTal1 mESC line | This paper | iTal1 |
| Runx1+/hCD4 | (*Sroczynska et al., 2009a*) | Runx1+/hCD4 |
| Mice | | |
| C57BL/6N | EMBL Rome | |
| *Gfi1b:GFP* knock-in mice | (*Vassen et al., 2007*) | |
| *Gfi1:GFP* knock-in mice | (*Yücel et al., 2004*) | |
| Recombinant DNA | | |
| Plasmid: p2lox_5TFs | This paper | p2lox_5TFs |
| Plasmid: p2lox_6TFs | This paper | p2lox_6TFs |
| Plasmid: p2lox_8TFs | This paper | p2lox_8TFs |
| Plasmid: p2lox_empty | (*Vargel et al., 2016*) | p2lox_empty |
| Plasmid: p2lox_K_Cbfb | This paper | p2lox_K_Cbfb |
| Plasmid: p2lox_K_Erg | This paper | p2lox_K_Erg |
| Plasmid: p2lox_K_Fli1 | This paper | p2lox_K_Fli1 |
| Plasmid: p2lox_K_Gata2 | This paper | p2lox_K_Gata2 |
| Plasmid: p2lox_K_Lmo2 | This paper | p2lox_K_Lmo2 |
| Plasmid: p2lox_K_Lyl1 | This paper | p2lox_K_Lyl1 |
| Plasmid: p2lox_K_Runx1 | This paper | p2lox_K_Runx1 |
| Plasmid: p2lox_K_Tal1 | This paper | p2lox_K_Tal1 |
| Plasmid: pGL4.10[luc2] | Promega | E6651 |
| Plasmid: pGL4.74[hRluc/TK] | Promega | E6921 |

*Continued on next page*

*Continued*

| Product or resource | Supplier | Reference N° |
|---|---|---|
| Software and algorithms | | |
| CellProfiler | (*Kamentsky et al., 2011*) | http://www.cellprofiler.org |
| CellSelect software | Wafergen | N/A |
| Fiji | (*Schindelin et al., 2012*) | http://fiji.sc/Fiji |
| FlowJo | Tree Star, Inc. | N/A |
| JMP software | SAS | N/A |
| Others | | |
| 96.96 dynamic array IFC | Fluidigm | BMK-M-96.96 |
| AATI Fragment Analyzer | AATI | DNF-474 |
| Ampure XP Beads | Beckmann Coulter | B23319 |
| BioAnalyzer | Agilent | N/A |
| Biomark HD system | Fluidigm | N/A |
| FACSAria | Becton Dickinson | N/A |
| FACSCanto | Becton Dickinson | N/A |
| Fluidigm C1 10–17 uM IFC chip | Fluidigm | 100–5760 |
| Fluorescence microscope for Wafergen chip imaging | Olympus | BX43F |
| Hard-Shell PCR plates | Bio-Rad | HSP9611 |
| HiSeq 2000 | Illumina | N/A |
| ICell8 System | Wafergen | N/A |
| IncuCyte HD | Essen Biosciences | N/A |
| MACS MicroBead Technology | Miltenyi Biotec | N/A |
| Moxi Z Mini Automated Cell Counter | ORFLO | MXZ001 |
| MultiSample NanoDispenser | Wafergen | N/A |
| NextSeq | Illumina | N/A |
| NuPAGE 12% Bis-Tris Protein Gels | Life Technologies | NP0343BOX |
| PCR barcoded indexes | Illumina | FC-131–200 |
| POLARstar Omega device | BMG Labtech | N/A |
| PROTRAN Nitrocellulose membrane | PerkinElmer | NBA085C |
| Qubit | Life Technologies | N/A |
| SmartChip Cycler | Bio-Rad | T-100 |
| Universal cDNA Reverse Trancribed by Random Hexamer: Mouse Normal Tissues | BioChain | C4334566-R |
| XCell SureLock Mini-Cell Electrophoresis System | Invitrogen | N/A |

## Acknowledgements

We thank Georges Lacaud (University of Manchester) for the Runx1[+/hCD4] ESC line; Michael Kyba (University of Minnesota) for the A2.lox.Cre ESC line; Tariq Möröy (Université de Montréal) for the *Gfi1:GFP* and *Gfi1b:GFP* mouse lines; Philip Hublitz (University of Oxford) for the design of the 8TF construct; Daniel Bilbao, Kalina Stantcheva and Cora Chadick (EMBL Rome FACS Facility) for cell sorting; Jelena Pistolic (EMBL Genomics Core Facility) for microarrays; Leo Chan and Mai Ho (Wafer-Gen) for sc-RNA-seq; Gang Sun (Fluidigm) for advice on the SINGuLAR Analysis toolset; and Imre Berger (University of Bristol) and Philip Avner (EMBL Rome) for fruitful scientific discussions. The European Molecular Biology Laboratory, the EU Marie Curie Actions Cofund and the Wellcome Trust supported this work.

## Additional information

### Funding

| Funder | Grant reference number | Author |
|---|---|---|
| EMBL Interdisciplinary Post-docs (EIPOD) Initiative | Post-doc fellowship | Isabelle Bergiers |

The funders had no role in study design, data collection and interpretation, or the decision to submit the work for publication.

### Author contributions

Isabelle Bergiers, Conceptualization, Formal analysis, Supervision, Investigation, Visualization, Methodology, Writing—original draft, Project administration, Writing—review and editing; Tallulah Andrews, Andreas Buness, Formal analysis, Visualization, Writing—review and editing; Özge Vargel Bölükbaşı, Investigation, Visualization, Writing—review and editing; Ewa Janosz, Natalia Lopez-Anguita, Kerstin Ganter, Kinga Kosim, Cemre Celen, Gülce Itır Perçin, Paul Collier, Bianka Baying, Investigation, Writing—review and editing; Vladimir Benes, Supervision, Investigation, Writing—review and editing; Martin Hemberg, Conceptualization, Formal analysis, Supervision, Funding acquisition, Writing—review and editing; Christophe Lancrin, Conceptualization, Formal analysis, Supervision, Funding acquisition, Investigation, Visualization, Methodology, Writing—original draft, Project administration, Writing—review and editing, Lead Contact

### Author ORCIDs

Isabelle Bergiers http://orcid.org/0000-0001-9622-7960
Tallulah Andrews https://orcid.org/0000-0003-1120-2196
Özge Vargel Bölükbaşı https://orcid.org/0000-0003-4013-6343
Christophe Lancrin http://orcid.org/0000-0003-0028-7374

### Ethics

Animal experimentation: All experiments were performed in accordance with the guidelines and regulations defined by the European and Italian legislations (Directive 2010/63/EU and DLGS 26/2014, respectively). All mice were bred and maintained at EMBL Rome Mouse Facility in accordance with European and Italian legislations (EU Directive 634/2010 and DLGS 26/2014, respectively).

### Decision letter and Author response

Decision letter https://doi.org/10.7554/eLife.29312.054
Author response https://doi.org/10.7554/eLife.29312.055

## Additional files

### Supplementary files

• Supplementary file 1. (Related to *Figures 1*, *3*, *4*, *5* and *8*.) Description of the genes used for single-cell quantitative RT-PCR. The genes are classified by categories. A gene can belong to several categories.
DOI: https://doi.org/10.7554/eLife.29312.031

• Supplementary file 2. (Related to *Figure 1*.) Single-cell quantitative RT-PCR data shown in *Figure 1A and B*. The first worksheet contains Ct values, the second log2 expression data and the third the metadata relative to the cells shown in *Figure 1A and B*.
DOI: https://doi.org/10.7554/eLife.29312.032

• Supplementary file 3. (Related to *Figure 1D*.) Single-cell quantitative RT-PCR data shown in *Figure 1D*. The first worksheet contains Ct values, the second log2 expression data and the third the metadata relative to the cells shown in *Figure 1D*.
DOI: https://doi.org/10.7554/eLife.29312.033

• Supplementary file 4. (Related to *Figure 3—figure supplement 2B*.) Single-cell quantitative RT-PCR data shown in *Figure 3—figure supplement 2B*. The first worksheet contains Ct values, the second log2 expression data and the third the metadata relative to the cells shown in the *Figure 3—figure supplement 2*.
DOI: https://doi.org/10.7554/eLife.29312.034

• Supplementary file 5. (Related to *Figure 3—figure supplements 3B* and *4A*,) Microarray data results shown in *Figure 3—figure supplement 3B*. The first worksheet contains metadata, the nine other sheets contain the results of expression contrast between two populations indicated in the name of the corresponding worksheet. The worksheets 10–14 contain information for the GO term analysis presented in *Figure 3—figure supplement 4A*.
DOI: https://doi.org/10.7554/eLife.29312.035

• Supplementary file 6. (Related to *Figure 4*.) Single-cell quantitative RT-PCR data shown in *Figure 4A and B*. The first worksheet contains Ct values, the second log2 expression data, the third the metadata relative to the cells shown in *Figure 4A and B*, the fourth the matrix for *Figure 4B* and the fifth the p-values for *Figure 4B*.
DOI: https://doi.org/10.7554/eLife.29312.036

• Supplementary file 7. (Related to *Figure 5*.) Single-cell quantitative RT-PCR data shown in *Figure 5C*. The first worksheet contains Ct values, the second log2 expression data and the third the metadata relative to the cells shown in *Figure 5C*.
DOI: https://doi.org/10.7554/eLife.29312.037

• Supplementary file 8. (Related to *Figure 5—figure supplement 2*.) Single-cell quantitative RT-PCR data shown in *Figure 5—figure supplement 2*. The first worksheet contains Ct values, the second log2 expression data and the third the metadata relative to the cells shown in *Figure 5—figure supplement 2*.
DOI: https://doi.org/10.7554/eLife.29312.038

• Supplementary file 9. (Related to *Figure 7*.) Results of the Spearman correlation analysis shown in *Figure 7B*. The first worksheet contains all correlation values shown in the heatmap of *Figure 7B*. The second contains the list of Fli1 target genes defined by *Wilson et al. (2010)*. The third contains the list of Runx1 target genes defined by *Wilson et al. (2010)*. The fourth contains the name of Fli1 targets among the 210 genes list and their corresponding Spearman correlation values. The fifth contains the name of Runx1 targets among the 210 genes list and their corresponding Spearman correlation values. The sixth worksheet contains the name of both Fli1 and Runx1 targets among the 210 genes list and their corresponding Spearman correlation values.
DOI: https://doi.org/10.7554/eLife.29312.039

• Supplementary file 10. (Related to *Figure 8* and *Figure 8—figure supplement 1*.) Single-cell quantitative RT-PCR data shown in *Figure 8E* and *Figure 8—figure supplement 1B*. The first worksheet contains Ct values, the second log2 expression data and the third the metadata relative to the cells shown in *Figure 8E*.
DOI: https://doi.org/10.7554/eLife.29312.040

• Supplementary file 11. (Related to all Figures except *Figure 7* and *Figure 7—figure supplement 1*.) Table summarizing the experiments done in the manuscript. For each figure panel, the type of experiments is described as well as the number of times they were carried out.
DOI: https://doi.org/10.7554/eLife.29312.041

• Transparent reporting form
DOI: https://doi.org/10.7554/eLife.29312.042

## Data availability
The following dataset was generated:

| Author(s) | Year | Dataset title | Dataset URL | Database and Identifier |
|---|---|---|---|---|
| Bergiers B, Andrews T, Vargel Bölükbaşı Ö, Buness A, Janosz E, Lopez-Anguita N, | 2017 | Single cell transcriptomics reveals new insights on the dynamical function of transcription factors during blood stem and progenitor cell formation | https://www.ncbi.nlm.nih.gov/geo/query/acc.cgi?&acc=GSE96986 | NCBI Gene Expression Omnibus, GSE96986 |

Ganter K, Kosim K,
Celen C, Itır Perçin
G, Collier P, Baying
B, Benes V, Hem-
berg M, Lancrin C

The following previously published datasets were used:

| Author(s) | Year | Dataset title | Dataset URL | Database and Identifier |
|---|---|---|---|---|
| Goode DK, Obier N, Vijayabaskar MS, Lie-A-Ling M, Lilly AJ, Hannah R, Lichtinger M, Batta K, Florkowska M, Patel R, Challinor M, Wallace K, Gilmour J, Assi SA, Cauchy P, Hoogenkamp M, Westhead DR, Lacaud G, Kouskoff V, Göttgens B, Bonifer C5 | 2016 | Comprehensive Epigenomic Analysis Reveals Dynamic Regulatory Programs Of Blood Development | https://www.ncbi.nlm.nih.gov/geo/query/acc.cgi?acc=GSE69101 | NCBI Gene Expression Omnibus, GSE69101 |
| Pereira CF, Chang B, Gomes A, Bernitz J, Papatsenko D, Niu X, Swiers G, Azzoni E, de Bruijn MF, Schaniel C, Lemischka IR, Moore KA | 2016 | Direct Conversion from Mouse Fibroblasts Informs the Identification of Hemogenic Precursor Cells In Vivo | https://www.ncbi.nlm.nih.gov/geo/query/acc.cgi?acc=GSE54574 | NCBI Gene Expression Omnibus, GSE54574 |

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
