## [Decision Letter]

Thank you for submitting your article "Single cell transcriptomics reveals a new dynamical function of transcription factors during embryonic hematopoiesis" for consideration by *eLife*. Your article has been reviewed by three peer reviewers, one of whom, Amy J Wagers (Reviewer #1), is a member of our Board of Reviewing Editors, and the evaluation has been overseen by Aviv Regev as the Senior Editor.

The reviewers have discussed the reviews with one another and the Reviewing Editor has drafted this decision to help you prepare a revised submission.

Summary:

In this manuscript, Bergiers et al. leverage the power of single cell gene expression analysis to demonstrate co-expression of heptad transcription factors (Erg, Fli1, Tal1, Lyl1, Lmo2, Runx1 and Gata2) in a subset of embryonic YS and AGM cells that appear to represent pre-HSPCs. They further document distinct patterns of expression of these and other selected genes in other embryonic populations at these anatomical sites at distinct developmental stages, ultimately suggesting the existence of multiple distinct populations with different lineage potential. Finally, applying clever systems for inducible overexpression of one or multiple of these transcription factors in an in vitro ES differentiation system, they demonstrate unique features of the gene expression networks controlled by these TFs, including unexpectedly opposing actions in the specification of endothelial versus pre-HSPC fates that depend upon the specific subset of factors expressed (notably involving a dual role of Fli1, which in collaboration with Erg supports a cell endothelial fate and in collaboration with Runx1/Gata2 supports a hematopoietic fate). They also described the transcriptional changes occurring in endothelial cells and vascular smooth muscle (VSM) cells upon overexpression of the heptad TFs (+Cbfb). Together, these results allow the authors to propose a new molecular model for EHT, and possibly other key developmental transitions that may invoke dual competing TF function. The systems and data described also may be of use in ongoing efforts to optimize the generation of definitive HSCs from pluripotent cell sources. Overall, the work presents new mechanistic insights into the process of blood formation and identifies a previously unrecognized transient population of progenitors in between endothelium and HSPCs. The work is likely to be of interest to a range of investigators studying development and hematopoiesis; however, there are several aspects of the work and presentation that require revision/modification.

Essential revisions:

1) The manuscript is written in a rather long-winded and confusing manner, such that the main message (Fli1 dual role) is highly diluted. In addition, the figure panels are very small and the text is difficult to read. The authors should edit the manuscript to improve the figure quality/design and flow of the narrative, and to present a more concise and clear analysis of the data and conclusions.

2) The assumption that the authors identified a pre-HSPC population (in AGM and yolk sac (YS)) based on the expression of hematopoietic and endothelial markers should be better justified. The authors claim that they produced a homogeneous population of pre-HSPCs (+8iTFs + Dox). However, this population does not completely cluster with the pre-HSPCs (detected in vivo in AGM and YS). Does it mean that a certain type of pre-HSPCs is produced in vitro (pro-HSCs, pre-HSCs type I or type II)? The authors should provide a more informative (phenotypic/functional) characterization of this population to support their conclusion.

3) The rationale for the analysis of the VSM population produced in vitro is unclear. This part of the results cuts the flow of the paper and is very confusing. The enriched VSM population is contaminated by 20% of endothelial cells, rendering the data difficult to interpret. Moreover, the eVSM population is never compared to the VSM cells from the AGM/YS. As VSM cells never give rise to HSCs in the AGM or YS in vivo, the authors should explain why the eVSM population is used to identify the gene regulatory network and not the endothelial cells?

4) It is not clear whether Erg and Fli1 (or similarly Gata2 and Runx1) have a redundant, complementary or synergistic roles. It would thus be interesting to test them separately (8iTFs without each TF and not without the two TFs at the same time); however, unless the authors have already generated the requisite hES lines, this is likely to require extensive additional studies. If such data are available, they should be added to the paper; however, if these experiments cannot be completed and added to the paper in a timely manner, the authors should instead comment/speculate on their possible individual or redundant roles.

5) The authors present several sets of sc-qRT-PCR data and use correlative analyses to make comparisons. For example, in Figure 1—figure supplement 1C and D, cell clusters are defined independently by hierarchical clustering (HC) in AGM and YS. The clusters are then aligned by correlation coefficients (Figure 1B) or in a PCA projection (Figure 1C) and the authors state they are similar. How would the HC analysis look if the data was analysed together? Ideally, an integrated analysis of all sc-qRT-PCR data (Figures 1-4) together should be added to give a more objective impression of similarities / differences.

6) In Figure 1D the authors use a low-dimensional space defined by the PCA of AGM and YS cells and map additional data (from *Gfi1^-/-^ Gfi1b^-/-^* cells) on top. They conclude from the location of these data in between Endo and pre-HSPC that the mutant cells were trapped in a transition state. In my experience, mapping data onto existing PCA spaces tends to map the data somewhere in the middle. How would this look if PCA was performed on all data? Which genes are responsible for the similarity and which ones fail to make the switch to pre-HSPC?

7) Why did the authors choose to include Cbfb with the heptad of TFs from Wilson et al. (Introduction section)? This seems kind of arbitrary and should be explicitly justified. Is this important for the results?

8) There seems to be a lot of fluctuation in the counts of the differentiation assays in Figure 4B, even in -dox (in which all cells should be the same). At the same time, the statistics on these data seem questionable. Why would the difference between 7=>16 (iFli1 VSM) be significant, but the difference 22=>0 (i8TFs EryPCs_II) not? Overall, the effects for the i8TFs seem striking enough compared to the rest to be convincing, but the more subtle differences between the single TF lines could be easily overlooked. The authors may need to increase the numbers on these data to make the conclusions more robust, and they also should check their statistics.

9) The authors use the PCA in Figure 4C to illustrate that i8TFs+dox make a very homogeneous population of pre-HSPCs. However, there seems to be a lot of variance not explained by the two shown dimensions. Why is that and does this bring into question the homogeneity of the population? Please provide additional text to explain/justify the conclusions drawn.

10) The focus on eVSM (subsection “Identification of gene regulatory network from single cell RNA sequencing data”) is confusing. Earlier they say that eVSM did not convert so well to pre-HSPC (Figure 4). Their microarray data shows more DEGs, but the induced cells from eVSM also had limited differentiation capacity (would that capacity be better if fully converted cells were FACS-purified first?). Is the limited efficiency in the induction in any way related to the fact that only a small percentage of the transcriptional variation is due to the difference between +dox and -dox (Figure 5C => PC2, variance explained = 1%)? Please add text to address/explain these points.

11) The GRN analyses are also rather confusing: Why were two networks used? The authors depict one (Figure 6D) but show descriptive metrics for another (Figure 6E, network not shown anywhere). This is a bit misleading. Why were additional genes used to infer the first, smaller network? Did it not work otherwise? How were the additional TFs chosen? What does the network really tell us? 26 of 145 target genes were found in the network. That's only 18%. Given that the network has over 7k genes in it, is this actually any more than expected by pure chance? In the end, the authors derive from the whole GRN part of the analysis the conclusion that there are subsets of TFs tilting the cell fate either toward hematopoiesis or toward endothelial fate (which is good and interesting), but it seems that maybe this could have been achieved by just looking at the correlation matrix in Figure 7B? Please add further text addressing these points and explaining/justifying the analytical workflow and output.

12) The authors deliberately chose a design to minimize batch effects (subsection “Identification of gene regulatory network from single cell RNA sequencing data”) for the C1 scRNA-seq data (which is great). However, they should also address in the text whether batch effects might in any way have affected other conclusions in the paper. Many runs of the PCR assays were performed; the authors should explain how they were balanced.

[Editors' note: further revisions were requested prior to acceptance, as described below.]

Thank you for resubmitting your work entitled "Single cell transcriptomics reveals a new dynamical function of transcription factors during embryonic hematopoiesis" for further consideration at *eLife*. Your revised article has been favorably evaluated by Aviv Regev (Senior editor), a Reviewing editor, and two reviewers.

The manuscript has been improved but there are some remaining issues that need to be addressed before acceptance, as outlined below.

In particular, while the manuscript has been greatly improved, there is a remaining confusion on the number of mice and independent experiments conducted in critical points in the paper, and on how a limited number of replicates in some places can be mitigated. Three items in particular need to be addressed:

1) The initial experiment (single cell qPCR) was based on a single litter replicate (one experiment, 4-6 embryos). This is partly mitigated by the follow ups, but needs to be stated clearly by discussing the shortcomings of the initial experiments. In addition, if any other data (including published data from other studies) can be highlighted as an external replicate it would be helpful as well, but is not strictly required.

2) Throughout the study, it is exceedingly difficult to follow the number of animals and independent experiments performed and thus to assess the strength of key conclusions. This is exacerbated because of the number of datasets and heterogeneity of technologies and experimental setups. We ask that the authors prepare an overview table listing clearly all the different samples used for the analyses in this paper. (We note that the meaning of "n=" is often quite obscure in the context of the paper).

3) The low numbers in the colony-forming assays is a particular concern. As it limits the reliability of the results presented in the respective part of the paper. To their credit, the authors are rather cautious about their claims about these assays. We appreciate that the comparison in the same assay enhances the validity, but ask that the authors note the samm number.

The full reviews are provided for context below.

Reviewer #1:

Bergiers and colleagues made great efforts to respond to my comments and concerns, and to improve the fluidity and understanding of the manuscript. Figures are more readable and better organized (although the readability of the gene names beside the heatmaps could be improved).

Major comments

1) A better identification of the pre-HSPC population generated by the authors was necessary. However, the fact that the population resembles to phenotypically defined pro-HSCs but to molecularly defined pre-HSCs type I reflect (to my opinion) the heterogeneity of the population generated. A functional analysis (e.g. transplantation, reaggregated in vitro culture) would have certainly better identified the pre-HSPC population. These experiments are time consuming but would have been of a great interest for the readers.

2) The part of the eVSM population was reduced and the indication that this population was used as a tool clarifies this part of the manuscript.

3) It would have been very interesting to test each TF separately (8iTFs without each TF and not without the two TFs at the same time). But discussing the potent redundancy or complementarity of these factors improved the discussion.

Finally, I was extremely surprised that in the revised version (where the number of experiments performed was specified), it was mentioned that only one experiment was performed by tissue and time point, using very few embryos (4 to 6 embryos). Three independent experiments are usually performed to avoid the risk of biased data.

Reviewer #2:

In brief, the revised manuscript submitted by Bergiers and colleagues has been markedly improved. I appreciate the additional analyses added and I can sort of accept the rationale / justification for starting experiment with eVSM, for arbitrarily including Cfcb, etc.

There are two main problems with this manuscript that still bother me:

1) Network analysis: The construction of the "small" network seems too arbitrary and the authors' rebuttal does little to convince me otherwise. Basically, the authors tried to build a network from the heptad of TFs (+Cfcb which they had arbitrarily added in before) and found it didn't give them any insights. So they looked at their favourite papers and added more nodes until they got an interesting result. How robust is this analysis? Can the same conclusions be reached by adding other TFs into the network? Do the conclusions hold if some nodes are removed? If those TFs are also considered of central importance to the processes studied, why were they not included in the previous experiments ("i12TFs")? To me, it would make a lot more sense and be much less biased, to build the "big" network (using a fast, but no-so-good correlation measure) first and to use this to inform which additional TFs to add to the small network (by using unbiased metrics calculated on the network, such as the. In their rebuttal, the authors state "the local network […] does not give a global picture of the network in the cell" – I agree, so what's the point of showing this at all? It's just to provide some nice-to-look-at (if you're into that kind of thing) network-y graphics that contains genes that ring interesting to people (because they were hand-picked from previous literature!). In a way, none of this would matter much if the network analysis was just a means to an end – i.e. to find the complementary TF networks biasing cells to different fates. However, the authors try to sell their network analysis approach as some sort of novel tool with wide applicability – which I just think it is not.

2) Colony-forming assays: The numbers here are weak and the authors pretty much admit that. I understand the experiments are difficult and time-consuming to perform and it wouldn't have been reasonable to ask for additional experiments. Still, I find these data as they stand somewhat unsatisfactory. Do the other reviewers find this convincing enough?

---

## [Author Response]

Essential revisions:1) The manuscript is written in a rather long-winded and confusing manner, such that the main message (Fli1 dual role) is highly diluted. In addition, the figure panels are very small and the text is difficult to read. The authors should edit the manuscript to improve the figure quality/design and flow of the narrative, and to present a more concise and clear analysis of the data and conclusions.

We corrected the manuscript and changed the figures to improve the flow of the narrative and the visibility of the figure panels. We combined the key data from Figures 1 and 2 together to help focus our message (see new Figure 1 and Figure 1—figure supplement 1 to 6). We have also edited significantly the corresponding sections of the text (see the new part “Key transcription factors identify a population intermediary between endothelial and blood cells during EHT in vivo and in vitro”).

Another major change corresponds to the eVSM section of the manuscript (former Figure 5 and former Figure 5—figure supplement 1, now in Figure 3—figure supplement 2, 3 and 4). We have trimmed significantly this part of the text to improve the flow in genera (see the part “Simultaneous overexpression of Erg, Fli1, Tal1, Lyl1, Lmo2, Runx1, Cbfb and Gata2 during hemangioblast differentiation leads to the formation of a population resembling Pre-HSPCs”).

2) The assumption that the authors identified a pre-HSPC population (in AGM and yolk sac (YS)) based on the expression of hematopoietic and endothelial markers should be better justified. The authors claim that they produced a homogeneous population of pre-HSPCs (+8iTFs + Dox). However, this population does not completely cluster with the pre-HSPCs (detected in vivo in AGM and YS). Does it mean that a certain type of pre-HSPCs is produced in vitro (pro-HSCs, pre-HSCs type I or type II)? The authors should provide a more informative (phenotypic/functional) characterization of this population to support their conclusion.

This is a very interesting question. To answer it, we have added experiments that are now presented in a part called “in vitro generated Pre-HSPCs share similarities with AGM Pre-HSCs type I”. See also the new Figure 5 and Figure 5—figure supplement 1 and 2. We characterized the population produced in vitro with the eight TFs and compared it to in vivo cells identified by the group of Alexander Medvinsky (Rybtsov et al., 2014). As shown in Figure 5A, the i8TFs+dox cells are VE-cad+CD45-CD41+CD43- similarly to the in vivo Pro-HSCs. However, a closer transcriptional characterization showed that transcriptionally the i8TFs+dox cells look like Pre-HSCs type I. This is in line with the functional data showing that i8TFs+dox cells are able to generate colonies in CFU-assay (Figure 4) like the Pre-HSCs type I while the Pro-HSCs have little CFU-C activity (see Figure 4 of Rybtsov et al., 2014).

3) The rationale for the analysis of the VSM population produced in vitro is unclear. This part of the results cuts the flow of the paper and is very confusing. The enriched VSM population is contaminated by 20% of endothelial cells, rendering the data difficult to interpret. Moreover, the eVSM population is never compared to the VSM cells from the AGM/YS. As VSM cells never give rise to HSCs in the AGM or YS in vivo, the authors should explain why the eVSM population is used to identify the gene regulatory network and not the endothelial cells?

We agree with the reviewers; the flow of our manuscript was slowed down because of the part on the eVSM population. We have drastically reduced this section as we explained in our response to the point 1.

As the reviewers pointed out correctly, the VSM cells do not generate HSC in vivo. That is why we did not isolate them from AGM and Yolk Sac. On the other hand, we did consider them in our in vitro model because the data obtained suggested that these cells were also changing dramatically. Indeed, at the moment of the dox treatment, they were very numerous in the BL-CFC culture (new Figure 3—figure supplement 2A) while after 48-hours treatment most of the culture is VE-Cad^+^ CD41^+^ suggesting that all the cells, including VSM cells, reacted to the treatment (new Figure 3A). We showed that it was indeed the case (new Figure 3—figure supplement 2C). We found that very interesting and thought that it emphasized the crucial function of the eight TFs.

Since our main goal was to use single-cell RNA sequencing to study the impact of the simultaneous expression of the eight TFs on the transcriptome, we found that VSM cells were very useful as a tool. Indeed, they express almost none of these TFs while the Endo cells express already Erg, Fli1, Lmo2 and Tal1. To understand how the 8TF induction by dox treatment affected GRNs, we needed to avoid as much as possible cells, which were already undergoing the EHT like the Endo population (see new Figure 3—figure supplement 2C). If we had used the Endo population for this analysis, we would have only a two-fold increase in VE-Cad^+^ CD41^+^ population between -dox and +dox conditions (43% versus 98%). On the other hand, the eVSM in no dox condition only gave rise to 9% of VE-Cad^+^ CD41^+^ in comparison to 88% in +dox condition. From this point of view, the eVSM was the most optimal solution because of this difference and also because of the gene expression difference detected by microarrays (new Figure 3—figure supplement 3B and new Supplementary File 5). Indeed, upregulated genes in eVSM+dox compared to eVSM-dox belonged to the “Immune system process” and “Vascular development” GO terms (new Figure 3—figure supplement 4A). This suggest the induction in these cells of a dual Endothelial/Hematopoietic identity compatible with the Pre-HSPCs population. Overall, these characteristics facilitated the development of the bioinformatics pipeline necessary to resolve the GRNs. It turned out to be the right approach as we identified Gpr56 as a key target of these TFs and the tools/approaches that we have developed with the eVSMs could be readily used with the Pereira dataset putting us on the track of the Runx1/Fli1 antagonistic function.

We have added text to explain better the reasons we used the eVSMs (see the first paragraph of the section called “Identification of gene regulatory network from single cell RNA sequencing data”).

4) It is not clear whether Erg and Fli1 (or similarly Gata2 and Runx1) have a redundant, complementary or synergistic roles. It would thus be interesting to test them separately (8iTFs without each TF and not without the two TFs at the same time); however, unless the authors have already generated the requisite hES lines, this is likely to require extensive additional studies. If such data are available, they should be added to the paper; however, if these experiments cannot be completed and added to the paper in a timely manner, the authors should instead comment/speculate on their possible individual or redundant roles.

This is a very interesting comment. Unfortunately, these four additional ESC lines have not been generated and it would have taken more than two months to do the required experiments. Based on sequence homology, we can hypothesize that some redundancy could exist between Erg and Fli1 that are both ETS transcription factors. Wilson and colleagues have also shown that in HPC7 more than 90% of the loci targeted by Fli1 are also targeted by Erg (Wilson et al., 2010). However, these loci represent only about 60% of the ones targeted by Erg, which might suggest a broader role for Erg. The possible redundancy between Erg and Fli1 is also supported by our single TF inducible cell line data. Flow cytometry and single-cell qRT-PCR analyses have shown that both TFs have similar effect on the differentiation (Figure 3—figure supplement 5D and Figure 4).

Regarding Runx1 and Gata2, our single TF inducible cell line data support their functional similarities (Figure 3—figure supplement 5D and Figure 4). Additionally, the network analysis suggests that Runx1 and Gata2 could cooperate together (Figure 7A). Indeed, they have both similar directional correlations with common targets. Finally, Wilson et al., showed also that Gata2 and Runx1 interact physically in vitro and showed that Runx1^+/−^::Gata2^+/−^ compound heterozygous embryos are not viable due to severe hematopoietic defects supporting the notion that the two proteins are involved in the same biological process. Since Gata2 and Runx1 are belonging to two different classes of transcription factors, it is more likely that they have complementary or synergistic role than being redundant.

Additional text has been added to the discussion in the section discussing our model (see new Figure 9).

5) The authors present several sets of sc-qRT-PCR data and use correlative analyses to make comparisons. For example, in Figure 1—figure supplement 1C and D, cell clusters are defined independently by hierarchical clustering (HC) in AGM and YS. The clusters are then aligned by correlation coefficients (Figure 1B) or in a PCA projection (Figure 1C) and the authors state they are similar. How would the HC analysis look if the data was analysed together? Ideally, an integrated analysis of all sc-qRT-PCR data (Figures 1-4) together should be added to give a more objective impression of similarities / differences.

As suggested, we analyzed all the sc-qRT-PCR together. We first combined the populations of AGM and YS to do a hierarchical clustering analysis (see Figure 1—figure supplement 4B). For both tissues, the groups shown in Figure 1 generally clustered according to their identity although some cells from the intermediary populations (AGM_SC_2 and YS_SC_2) clustered with the Endothelial groups (AGM_SC_1 and YS_SC_1), possibly because of a higher endothelial gene expression.

Moreover, we combined the single-cell qRT-PCR results of a total of 1,660 cells (Figures 1, 4 and 5) and compared them to each other using PCA (see Figure 5—figure supplement 2). As shown in this PCA analysis, the combined analysis of all the sc-qRT-PCR data confirmed what had been stated in the first figures of this manuscript. The expression of the selected 95 genes is sufficient to follow the endothelial to hematopoietic transition. Cells from in vivo AGM and yolk sac, together with in vitro cells from the inducible lines, are spread along the transition without major clustering due to the origin of the cells (in vivo, in vitro, embryonic regions, etc). The clusters identified based on the individual analyses presented in the Figure 1, 4 and 5 can be located on the PCA and fit with our previous observations. Vascular smooth muscle cells from in vitro analyses are clustering apart. Endothelial clusters from AGM, YS and in vitro samples are all in the same part of the PCA plot, connected to hematopoietic progenitors through the Pre-HSPCs from in vivo and in vitro samples and Pre-HSCs from in vivo. As expected, Pro-HSCs are located closer to the endothelial population while the Pre-HSCs type I are closer to the hematopoietic progenitors.

6) In Figure 1D the authors use a low-dimensional space defined by the PCA of AGM and YS cells and map additional data (from Gfi1^-/-^ Gfi1b^-/-^ cells) on top. They conclude from the location of these data in between Endo and pre-HSPC that the mutant cells were trapped in a transition state. In my experience, mapping data onto existing PCA spaces tends to map the data somewhere in the middle. How would this look if PCA was performed on all data? Which genes are responsible for the similarity and which ones fail to make the switch to pre-HSPC?

We substituted the PCA projection analyses of former Figure 1D by an integrated analysis to have a more objective impression of similarities and differences. As shown in new Figure 1B, the conclusions drawn are similar to the ones based on the PCA projection analysis. The only difference is that the *Gfi1^-/-^Gfi1b^-/-^* cells are clustered more with the Pre-HSPC group. In general, there is no major gene expression difference between these groups apart for the absence of Gfi1 and Gfi1b expression. However, *Gfi1^-/-^ Gfi1b^-/-^* cells express more the endothelial genes Pcdh12 and Sox7 and less the hematopoietic genes Ptprc, Sla, Mpo, Itgam, Myb and Spi1 (see new Figure 1E).

7) Why did the authors choose to include Cbfb with the heptad of TFs from Wilson et al. (Introduction section)? This seems kind of arbitrary and should be explicitly justified. Is this important for the results?

Cbfb is the β subunit of the heterodimeric complex in which Runx1 is the α subunit. The function of Cbfb is to enhance Runx1 binding to the DNA. Moreover, Cbfb has been shown to be essential for Runx1 function in vivo (Wang et al., 1996; Tahirov et al., 2001). Based on these elements, we considered both proteins to be a single functional unit and that is why Cbfb was included in our polycistronic transgenic construct. That way we were able to keep the ratio between the two partners while overexpressing the transcription factors upon doxycycline treatment making sure that Runx1 could work properly.

8) There seems to be a lot of fluctuation in the counts of the differentiation assays in Figure 4B, even in -dox (in which all cells should be the same). At the same time, the statistics on these data seem questionable. Why would the difference between 7=>16 (iFli1 VSM) be significant, but the difference 22=>0 (i8TFs EryPCs_II) not? Overall, the effects for the i8TFs seem striking enough compared to the rest to be convincing, but the more subtle differences between the single TF lines could be easily overlooked. The authors may need to increase the numbers on these data to make the conclusions more robust, and they also should check their statistics.

The reviewers are right, increasing the number of cells would help to control the fluctuations in the differentiation assay and offer more power to detect changes. Unfortunately, due to lack of time, we could not do it. However, we have improved our statistical analysis, added a new visualization and provided additional plots aiming to support our conclusions.

Our analysis is based on the cell counts as shown in Figure 4B but also accounts for the total number of successfully classified cells per cell line (sum of all cells per cell line in Figure 4B). To better depict this aspect, we generated an additional visualization that displays all calibrated data (“fractions of cells”), rearranged by pooling all “-dox” controls to emphasize the contrasts of interest. We also added a boxplot (Author response image 1) indicating the variation within the “-dox” control groups. Each individual “+dox” data point is compared to its respective “-dox” control group (Figure 4—figure supplement 1).

In contrast to the model used in the manuscript, if our results were analysed by a standard linear model (with Gaussian distribution) followed by multiple testing correction, then indeed the 22=>0 (i8TFs EryPCs_II) change would be significant while the 7=>16 (iFli1 VSM) would not remain significant as indicated in Author response image 1. This is in line with the reviewers´ comments.

However, this model assumes that the data are continuous and this is not valid for data based on discrete counts like ours. So, we decided to choose a generalized linear model, initially with Poisson distribution (original submission) and now with quasi-Poisson distribution. Indeed, the model with quasi-Poisson distribution appears to be more stringent with our data (Author response image 2). Results are indicated in Author response image 2 and were added in Figure 4—figure supplement 1. There are five significant contrasts compared to seven as reported in the previous version of our manuscript (Figure 4).

**Author response image 2. respfig2:** 

Still, these results should be taken with precaution since the high number of singletons challenges the model-fitting algorithm. Besides that, the statistical model does not fully address the multivariate nature of the data. Nonetheless, in line with the additional visualization, we consider our revised results relevant for the current manuscript and the prioritization of the follow up work described in the new Figure 8 (old Figure 7).

9) The authors use the PCA in Figure 4C to illustrate that i8TFs+dox make a very homogeneous population of pre-HSPCs. However, there seems to be a lot of variance not explained by the two shown dimensions. Why is that and does this bring into question the homogeneity of the population? Please provide additional text to explain/justify the conclusions drawn.

We agree with the reviewers’ observation that the variance explained by the first two principal components as shown in Figure 4C (now Figure 4—figure supplement 2B) is comparably low. However, if the PCA is calculated for the top 100 genes of highest variance instead of the top 400 as in Figure 4—figure supplement 2B then the explained variance increases to 30% (PC1+PC2, see Author response image 3).

**Author response image 3. respfig3:** 

Moreover, based on the PC2 variance, we noticed that there is less variance for the i8TF+dox cells compared to the Empty-dox and Empty+dox cells. This would suggest that the i8TF+dox cells are comparatively more homogeneous than the control cells. In general, the single cell RNA-seq tends to be less sensitive (fewer genes detected) and less precise (higher technical noise) when compared to bulk sequencing data. Thus, the variance explained in the PCA plot as obtained with this technology may not match expectations built upon standard bulk RNA-sequencing data

10) The focus on eVSM (subsection “Identification of gene regulatory network from single cell RNA sequencing data”) is confusing. Earlier they say that eVSM did not convert so well to pre-HSPC (Figure 4). Their microarray data shows more DEGs, but the induced cells from eVSM also had limited differentiation capacity (would that capacity be better if fully converted cells were FACS-purified first?). Is the limited efficiency in the induction in any way related to the fact that only a small percentage of the transcriptional variation is due to the difference between +dox and -dox (Figure 5C => PC2, variance explained = 1%)? Please add text to address/explain these points.

For an explanation of our focus on the eVSM population, please see point 3. However, in order to understand why these cells were not able to give rise to more blood colonies in the CFU-assay, we re-examined our microarray data and compared eVSM+dox and Endo+dox (see contrast VSMact_Endoact worksheet in Supplementary File 5). We found genes, which could potentially be responsible for the incapacity of the eVSM+dox to generate blood colonies. Tgfb2 an inhibitor of the EHT (Vargel et al., 2016), the VSM markers Pdgfrb, Tagln and Acta2 were all expressed at a higher level in eVSM+dox while Csf3r, coding for the receptor of the hematopoietic growth factor G-CSF was downregulated in eVSM+dox. Among all these genes, we believe that the expression of Tgfb2 could be the main reason why there were few colonies formed from eVSM+dox cells. The expression of VSM markers could be indicative of the fact that the VSM program has not been fully repressed which could also explain why few blood colonies were generated. The list of fold change is below (positive values indicate higher expression in eVSM while the negative one indicates lower expression):

SymbolEnsembl_IDLog_2_FoldChangeAverage ExpressionAdjusted P.ValTgfb2ENSMUSG000000392392.437.821.74E-06Acta2ENSMUSG00000035783211.266.82E-06TaglnENSMUSG000000320851.3110.347.73E-05PdgfrbENSMUSG000000246201.126.670.000909569Csf3rENSMUSG00000028859-1.174.310,000645884

Finally, we agree with the reviewers, the FACS-purification of the eVSM+dox cells, in particular of the cKit+ ones, may potentially lead to a better output in the CFU-C assay.

11) The GRN analyses are also rather confusing: Why were two networks used? The authors depict one (Figure 6D) but show descriptive metrics for another (Figure 6E, network not shown anywhere). This is a bit misleading. Why were additional genes used to infer the first, smaller network? Did it not work otherwise?

In Figure 6D (now Figure 6A), the first network (based on dCor) is only the local network of the 12 seed genes, it does not give a global picture of the network in the cell – i.e. only correlations between those 12 genes and all others were considered (~240,000 total correlations). The seed genes would by construction be the most important in the local network. However, because dCor is very slow it is computationally infeasible to calculate the global network with it.

In contrast, the second network in Figure 6E (now Figure 6B) is the global network – i.e. the correlation between all possible pairs of genes was calculated – so it makes sense to look at how important the seed genes are in the whole network. However, to calculate the correlation for all 142 million possible pairs, we had to use a very fast correlation method thus we used Spearman correlation.

How were the additional TFs chosen?

With only the eight TFs as seeds, the small network for the sc-RNA-Seq data on the i8TFs cells was much smaller and did not contain either Runx1 and Fli1 (Author response image 4 left panel). Since these factors are well established as important in EHT and hematopoiesis, we hypothesized that these factors may participate in more complex relationships through other TFs we had not initially considered. Thus, we considered several prominent papers (Beck et al., 2013; Moignard et al., 2015; Tanaka et al., 2012; Bonzanni et al., 2013; Moignard et al., 2013) examining these processes and added additional TFs which were included in the network in at least two papers by different authors. For consistency, all 13 TFs were used in the Pereira data as well, however the inverse relationship between Fli1 and Runx1 apparent in this network does not depend on the inclusion of the additional TFs (see Author response image 4 right panel). The text has been updated to clarify this point.

**Author response image 4. respfig4:** (Left) 8TFs network in our sc-RNA-Seq data. (Right) 8TFs network in Pereira sc-RNA-Seq data.

What does the network really tell us? 26 of 145 target genes were found in the network. That's only 18%. Given that the network has over 7k genes in it, is this actually any more than expected by pure chance?

No, this was not more than expected by chance, p= 0.3 using a hypergeometric test. This has been noted in the text.

In the end, the authors derive from the whole GRN part of the analysis the conclusion that there are subsets of TFs tilting the cell fate either toward hematopoiesis or toward endothelial fate (which is good and interesting), but it seems that maybe this could have been achieved by just looking at the correlation matrix in Figure 7B? Please add further text addressing these points and explaining/justifying the analytical workflow and output.

We focused on the subsets of TFs tilting cell fates towards hematopoiesis/endothelial fate as this was the most prominent and robust finding of our work. However, identifying this relationship within a large gene-gene correlation matrix is not trivial due to the large number of other correlations present and its relatively hard to interpret “hairball” structure (see Author response image 5), which is also the reason why a visualization of the full gene-gene correlation network was not included in the original manuscript.

**Author response image 5. respfig5:** 

Thus, we performed targeted analysis of the key TFs and their interactions, which also revealed other interesting potential relationships – such as the role of Gata2 and Lmo2 in regulating different subsets of the genes involved in the Fli1-Runx1 antagonism (Figure 7A), or the core network of Lmo2-Erg-Lyl1 (Figure 6A).

12) The authors deliberately chose a design to minimize batch effects (subsection “Identification of gene regulatory network from single cell RNA sequencing data”) for the C1 scRNA-seq data (which is great). However, they should also address in the text whether batch effects might in any way have affected other conclusions in the paper. Many runs of the PCR assays were performed; the authors should explain how they were balanced.

To control for technical variation in our sc-qRT-PCR, we used a commercial cDNA mix allowing us to detect consistently most of our genes of interest (93 genes out of 95). In the bar graph in Author response image 6 we have plotted the average Ct Values of the 93 genes from 19 independent PCR runs with the Fluidigm Biomark HD. The standard deviation is indicated as an error bar in the graph.

**Author response image 6. respfig6:** 

As shown in this plot, we had very little variation in performance across the different runs (average standard deviation 0.3). This is consistent with the results of our analysis where we showed that the data from Figures 1, 4 and 5 could be combined in a PCA plot and cells with similar characteristics were clustering with each other even when they were coming from different experiments (Figure 5 —figure supplement 2).

We have now mentioned in the text how we controlled for technical noise in our sc-qRT-PCR (see the “Single-cell quantitative RT-PCR” section in the Material and Methods).

[Editors' note: further revisions were requested prior to acceptance, as described below.]

In particular, while the manuscript has been greatly improved, there is a remaining confusion on the number of mice and independent experiments conducted in critical points in the paper, and on how a limited number of replicates in some places can be mitigated. Three items in particular need to be addressed:1) The initial experiment (single cell qPCR) was based on a single litter replicate (one experiment, 4-6 embryos). This is partly mitigated by the follow ups, but needs to be stated clearly by discussing the shortcomings of the initial experiments. In addition, if any other data (including published data from other studies) can be highlighted as an external replicate it would be helpful as well, but is not strictly required.

We understand the reviewers’ concerns. After checking carefully our notes, we found out that the number of embryos stated previously (4-6) was not correct. We apologize for our error. The right numbers are listed below. We also added in the Materials and methods section the procedure that we used to isolate AGM and yolk sacs.

**Time point****Number of litter****Total number of used embryos****Somite pairs range****E9**1714 to 21**E10.5**213 (7 + 6)33 to 37**E11**1939 to 43

The Figure 1A is the result of three runs of single-cell q-RT-PCR performed independently (each run corresponded to one time point). For each time point, AGM and Yolk sac cells always came from the same litter of mouse embryos. The single-cell FACS sorting of AGM and Yolk Sac cells was always done on the same day for each time point (E9, E10.5 and E11).

Even though most of our results were based on single litter replicate, we are confident that our conclusions are correct because similar populations were found at different time points (hence derived from different litters) and were shared between AGM and Yolk sac. Indeed, hierarchical clustering analysis showed that the Endo and Pre-HSPC groups are composed of cells coming from all time points (four different litters) and all tissues while HSPCs were found at E10.5 and E11 (three different litters) and from both AGM and Yolk sac.

Moreover, the group of Niels-Bjarne Woods has recently identified populations similar to ours using single-cell q-RT-PCR and the human in vitro Endothelial-Hematopoietic transition (EHT) model. This model is based on the differentiation of human ES cells (Guibentif C et al., 2017). In particular, they found that the intermediate population between endothelial and blood cells during EHT (called Pre-HSPC in our study and EHT cluster in the Guibentif publication) co-expresses at the single-cell level Runx1, Lyl1, Tal1, Gata2, Erg and Fli1 (Lmo2 was not assessed) similarly to the Pre-HSPC population that we described (see Figure 1D of Guibentif C et al.). Although Guibentif C et al. did not study further the functional meaning of the co-expression of these transcription factors; their results reinforce our conclusions and suggest the evolutionary conservation of the EHT process (see Discussion, second paragraph).

2) Throughout the study, it is exceedingly difficult to follow the number of animals and independent experiments performed and thus to assess the strength of key conclusions. This is exacerbated because of the number of datasets and heterogeneity of technologies and experimental setups. We ask that the authors prepare an overview table listing clearly all the different samples used for the analyses in this paper. (We note that the meaning of "n=" is often quite obscure in the context of the paper).

As requested, we have now added an overview table listing clearly the different samples used for the analyses in our paper. Please see supplementary file 11.

3) The low numbers in the colony-forming assays is a particular concern. As it limits the reliability of the results presented in the respective part of the paper. To their credit, the authors are rather cautious about their claims about these assays. We appreciate that the comparison in the same assay enhances the validity, but ask that the authors note the samm number.

We agree with the reviewers, the frequency of cells (i.d. CFU/number of cells plated x100) giving rise to blood colonies is low. The maximum frequency we have got is around 2% (Figure 3E, CFU-Mac+dox). Nonetheless, our results are in line with the performance of this type of assay in previous publications. Going through a series of papers using the same protocol of CFU-assay and cells derived from differentiated mouse ESC, we found that the frequency of cells giving rise to blood colonies ranged from 0.033% and 8% (see table below). In that context, the frequencies that we have obtained are within the range of related published works using similar cells and protocol.

In addition, we would like to stress that the frequencies we obtained were consistent across several independent experiments and we showed reliably that the overexpression of the 8 TFs increased the frequency of CFU in a given population (Figure 3E and Figure 3—figure supplement 2D).

To facilitate the comparison of the CFU-assay results between Figure 3E and Figure 3—figure supplement 2D, we are now showing the same number of plated cells in both graphs (i.d. 35 000 cells, see new Figure 3—figure supplement 2D).

**Publication****Figure Number****Type of plated cells in the CFU-assay****Minimum CFU Frequency (CFU/number of cells plated x 100)**
**Maximum CFU Frequency (CFU/number of cells plated x 100)****Ferreras C et al., 2011 (PMID: 21498670)**1FPurified CD41^+^ cells from differentiated mouse ESC0.1253**Boros K et al., 2011 (PMID: 21782766)**3GPurified CD41^+^ cells from differentiated mouse ESC0.150.84DPurified CD41^+^CD34^+^ cells from differentiated mouse ESC18**Nasrallah R et al., 2015 (PMID: 25979706)**2ECells from Day 2 BL-CFC culture (derived from differentiated mouse ESC)0.0830.833BCells from Day 2 BL-CFC culture (derived from differentiated mouse ESC)0.0330.46**Pearson S et al., 2015 (PMID: 25660408)**2CCells from Day 2 BL-CFC culture (derived from differentiated mouse ESC)354BSorted c-Kit^+^ cells from Day 2 BL-CFC culture1.78**Garcia-Alegria E et al. 2016 (PMID: 27197878)**2ECells from BL-CFC culture (derived from differentiated mouse ESC)1.54